# The Grand Illusion: The Myth of Software Portability and Implications for ML Progress.

**Fraser Mince**[*]
Cohere for AI Community
`frasermince@gmail.com`

**Dzung Dinh**[*]
Cohere for AI Community
`dinhd@dickinson.edu`

**Jonas Kgomo**
Cohere for AI Community
`jonaskgmoo@gmail.com`

**Neil Thompson**
MIT
`neil_t@mit.edu`

**Sara Hooker**
Cohere for AI
`sarahooker@cohere.com`

## Abstract

Pushing the boundaries of machine learning often requires exploring different hardware and software combinations. However, the freedom to experiment across different tooling stacks can be at odds with the drive for efficiency, which has produced increasingly specialized AI hardware and incentivized consolidation around a narrow set of ML frameworks. Exploratory research can be restricted if software and hardware are co-evolving, making it even harder to stray away from mainstream ideas that work well with popular tooling stacks. While this friction increasingly impacts the rate of innovation in machine learning, to our knowledge the lack of portability in tooling has not been quantified. In this work, we ask: *How portable are popular ML software frameworks?* We conduct a large-scale study of the portability of mainstream ML frameworks across different hardware types. Our findings paint an uncomfortable picture – frameworks can lose more than $40\%$ of their key functions when ported to other hardware. Worse, even when functions are portable, the slowdown in their performance can be extreme and render performance untenable. Collectively, our results reveal how costly straying from a narrow set of hardware-software combinations can be - and suggest that specialization of hardware impedes innovation in machine learning research.

## 1 Introduction

The field of machine learning (ML) has made significant strides in recent years, thanks in large part to advances in hardware and software [Chowdhery et al., 2022, Zhang et al., 2022, Kaplan et al., 2020]. However, the pursuit of efficiency has led to the creation of increasingly specialized AI hardware and the consolidation of ML frameworks around a narrow set of tools [Hooker, 2021]. This specialization has limited the ability of researchers to experiment with different hardware and software combinations, hindering the rate of innovation in the field.

The portability challenge has been amplified by the ever more heterogeneous landscape of hardware and software [Reddi et al., 2020]. In particular, differences in hardware create a vexing problem for software: how to allow portability while maximizing performance [Hooker, 2021, Lee et al., 2011, Barham and Isard, 2019]. Many commercial hardware suppliers purport to support a variety of popular ML libraries, however qualitative evidence from machine learning researchers suggests that this is often far from a straightforward process that requires significant changes to the code before it

---

[*]These authors contributed equally to this work.

37th Conference on Neural Information Processing Systems (NeurIPS 2023).

can be transferred successfully [Johansen et al., 2014]. In this work, we ask how has the *increasingly fragmented and specialized hardware and software landscape impacted the portability of research?*

To our knowledge, there has been no prior work that has sought to quantify the ease of portability between hardware types. In this work, we seek to address this gap, by explicitly quantifying the portability of popular mainstream ML libraries, TensorFlow [Abadi et al., 2015], PyTorch [Paszke et al., 2019], and JAX [Bradbury et al., 2018], that are used by millions of developers across different hardware types. We embark on extensive data collection and annotation to hand-curate representative tests for each library and subsequently benchmark transferability and latency across different hardware types.

Our results reveal highly uneven portability, suggesting that there will be increasingly uneven gains from progress in computer science research. Exploration in ML research appears to be hindered by failing functions and dismal performance. While some operations benefit from portability across devices, there are large gaps in coverage for widely used software frameworks. We find that there are frustrating differences in the subset of software operations supported on different types of hardware which prevent the portability of algorithms across hardware types. Even where there is portability, significant gaps exist in performance between each framework. Software kernels are often overly optimized for a specific type of hardware which causes huge lags in efficiency when used with a different type of hardware [Hennessy and Patterson, 2019b]. Our main contributions can be enumerated as follows:

- We gather a human-curated and annotated collection of functions from popular ML libraries that can be benchmarked across hardware types. We open source this dataset for use in future benchmarking at the provided repo: `https://github.com/for-ai/portability`.

- We find that PyTorch and TensorFlow, in particular, have portability issues. On GPUs, 22% of the TensorFlow benchmark functions fail partially or completely. On TPUs, a remarkable 44% of PyTorch benchmark functions partially or completely fail.

- Even where functions are portable, we see significant gaps in performance – with both unexpected speedups and slowdowns moving functions between the GPU and the TPU. For example, 81.4% of functions in PyTorch exhibit more than a 10x slowdown when transferring functions from GPU to TPU.

- We illustrate that certain software libraries are locked into a particular tooling stack. JAX was co-designed with TPUs in mind, and this is reflected in its performance. In all, 91.8% of our JAX function set is faster on the TPU.

- We compare how software portability has evolved over time by comparing different versions of GPUs and TPUs. Specifically, we run experiments on both GPUs T4 and A100 and observe that the portability remains the same for PyTorch while it differs by only up to 1% for TensorFlow and JAX. Moreover, we observe that 28.07% and 9.09% of PyTorch functions achieve a 1.5X speed improvement when operating newer GPU and TPU versions, respectively. Hence, although newer generations of hardware have not improved software portability, they have yielded modest speed enhancements for certain frameworks.

**Importance of this work:** This paper presents an evaluation framework at the beginning of a time when hardware and software specialization is growing, and thus where comparative evaluations will become ever more important. The economics of chip specialization have dramatically changed over the last decade or so [Thompson and Spanuth, 2021], leading Hennessy and Patterson to term this a *new golden age for computer architecture* in their Turing lecture [Hennessy and Patterson, 2019a]. Specialization carries with it radical changes in performance, and disparities will only increase, as will the importance of co-designing implementations to those chips. Thus, we should expect that the type of quantitative portability analyses that we do in this paper will only become more important in the coming years to aid the design of tooling that is both efficient and portable.

## 2   Methodology

We are interested in quantifying the portability of mainstream Python libraries used for machine learning workloads. We define portability as the *ease with which a machine learning workload (code, data, and models) can transfer between different hardware types*. We consider several types of failure:

| Comparison of TPU and GPU Failure and Success Rates | | | | | | |
| --- | --- | --- | --- | --- | --- | --- |
| | GPUs | | | TPUs | | |
| | Success | Failure | | Success | Failure | |
| | Pass | Partial | Complete | Pass | Partial | Complete |
| TensorFlow | 78% | 8% | 14% | 71% | 15% | 14% |
| PyTorch | 92% | 3% | 5% | 57% | 27% | 17% |
| JAX | 98% | 0% | 2% | 97% | 0% | 3% |

Table 1: Comparison of portability success and failure rates of a random stratified sample of Tensor-Flow, PyTorch, and JAX functions across TPUs and GPUs.

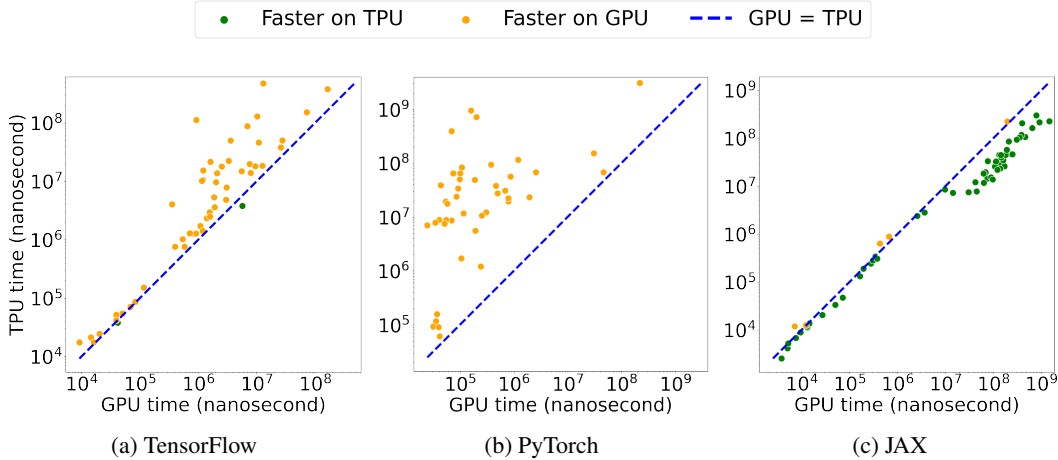

|  (a) TensorFlow | (b) PyTorch | (c) JAX |
| --- | --- | --- |

Figure 1: Comparison of average execution time on Log scale for TensorFlow, PyTorch, and JAX functions on GPU versus TPU. In total, there are 51 functions in TensorFlow, 43 functions in PyTorch, and 61 functions in JAX. The number of data points is lower than the overall count of functions because we have excluded all subtests that failed on either device. This exclusion was to ensure a valid comparison.
.

1. **Complete failure to run**: If the function does not run on the device at all.

2. **Partial failure to run**: Some but not all the benchmark tests for a given function fail to run.

3. **Intolerable latency**: High latencies may be prohibitively inefficient, which may impair usability even if the function technically is able to run on multiple hardware types.

Our goal is to benchmark the portability of libraries that claim to be portable across hardware types, and which are widely adopted. Hence, we evaluate the portability of JAX version 0.4.8, PyTorch version 1.12.0, and TensorFlow version 2.11.0.

## 2.1 Data collection

**Function sampling procedure**: To obtain a full list of all functions, we iterate through the module structure of PyTorch, TensorFlow, and JAX to enumerate all functions and classes contained in the library. This process results in 2718 TensorFlow functions, 2898 PyTorch functions, and 1091 JAX functions.

**Sampling procedure**: To have a representative view of each libraries performance, we do stratified sampling, including 1) **the top 20** functions as ranked by frequency of use, and 2) **5 random functions** from each decile of all functions ranked by frequency of use for each library (JAX, PyTorch, TensorFlow). The random sample allows us to capture a variety of different engineering

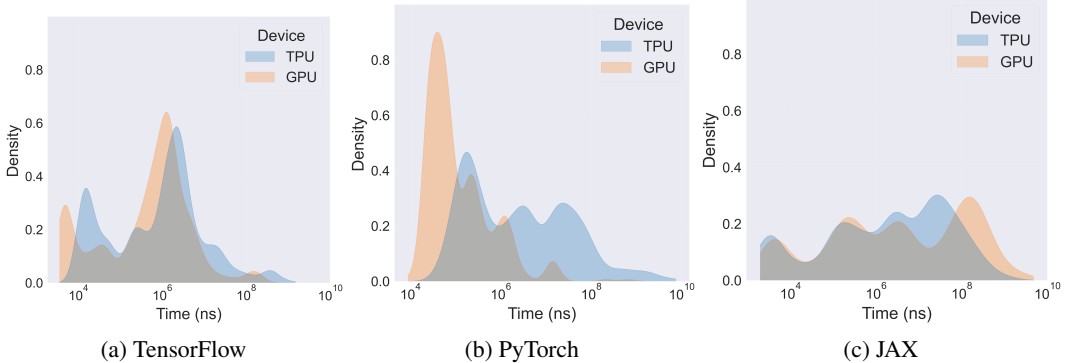

Figure 2: TensorFlow, PyTorch, and JAX time densities.

use cases and not overfit to scripts that may only rely on a small subset of the library. Benchmarking the top 20 functions measures how the frequency of use of a given function impacts portability – our expectation at the outset was that more frequently used functions would be prioritized for support across hardware types.

To identify the top 20 functions and the decile samples, we measure the frequency of how often these PyTorch, TensorFlow, and JAX functions appear in scripts in the CodeParrot-clean dataset[2]. CodeParrot-clean is a deduplicated version of the CodeParrot dataset[3], which is a collection of approximately 22 million Python files used originally to build a code generation model as part of the O'Reilly Transformers book [Tunstall et al., 2022]. This was created by collecting Python files from the Github Google BigQuery dataset [4]. We filtered to restrict to files that string matched import statements from either library. In the Appendix Section 12, we provide more details about the filtering procedure used.

Thus, for each framework, we sample approximately 70 functions, 50 random decile functions, and the 20 most-used.[5] The total number of samples per framework balanced the need for coverage with the time-intensive process need to human annotate and procure the minimal test for each function, to add in code to track latency to each script, and to modify and verify that each test is only run on the hardware in question. We intend to release this as a dataset that can be used by others for benchmarking portability.

**Human annotation and curation of tests**: Our goal is to benchmark as conservatively as possible the expected behavior of the function. To do so, we rely where possible on the test included in the library for a given function. We manually match each function to its associated tests in the TensorFlow, PyTorch, and JAX libraries. Given these are widely used and well-maintained libraries, our expectation is that the tests within the formal library reasonably represent the minimal code needed to validate expected function behavior under different conditions. For all tests that failed, we ensured that all failed tests were due to errors in the function being tested. Once tests are identified, we manually modify the test files to ensure 1) only the relevant tests and the initialization code needed for running were preserved, 2) the code was being run on the device we were evaluating, and 3) the code needed to record the latency in a consistent way for each script was added.

**Top 20 test exclusion:** In the top 20 functions, there were occasions when it was not possible to test a function:

1. **Overlapping functions**: Due to the inherited randomness of our sampling and the static nature of our top 20 there are some overlaps between deciles and the overall top 20: 4, 0, and

---

[2]https://huggingface.co/datasets/codeparrot/codeparrot-clean

[3]https://huggingface.co/datasets/transformersbook/codeparrot

[4]https://cloud.google.com/blog/topics/public-datasets/github-on-bigquery-analyze-all-the-open-source-code

[5]This number is approximate due to needing to exclude some functions from the list. More details are in the next paragraph. In all, we include 63 unique functions from PyTorch, 65 unique functions from TensorFlow, and 63 functions for JAX.

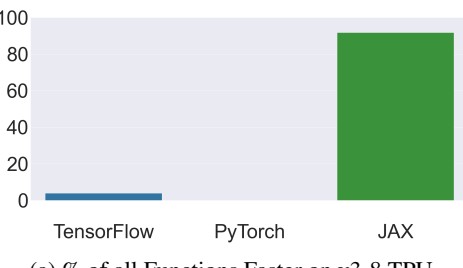
(a) % of all Functions Faster on v3-8 TPU

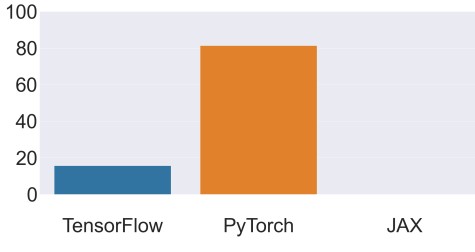
(b) % of all Functions 10X Faster on A100 GPU

Figure 3: Percentage of functions faster on A100 GPU vs v3-8 TPU.
.

4 overlapping functions in PyTorch, TensorFlow, and JAX, respectively. These we excluded from testing since we could not replace the top 20.

2. **Functions without a relevant test**: For some functions in the top 20 there was either no relevant test or situations where testing them would be somewhat nonsensical such as `int`, which shows up in PyTorch but as it is a type. It is not quite clear what we should test in this case, and thus we decided to exclude them.

**Replacement criteria**: After completing the sampling procedure, we found that a subset of functions was not viable for benchmarking. In total, we found 14 functions in TensorFlow, 13 functions in PyTorch, and 13 functions in JAX that needed replacement by resampling at random from the same decile. Our criteria for justifying replacing a function are detailed below:

1. No test present in the respective test suites. For example, `arctan` within PyTorch was not tested in the PyTorch open-sourced test suite. Respectively, there were 12, 12, and 13 functions for PyTorch, TensorFlow, and JAX that were not tested in the open-sourced test suite.

2. The tests are designed to validate the error handling of the functions; therefore, the timing doesn't work in this case. For example, when testing the `batch_norm` function in PyTorch, the test focuses solely on checking if the function correctly raises errors for unreasonable inputs. So, the core functionality of the method is not tested, just error throwing. Only one function in PyTorch fell into this case.

3. The functions operate on an older version of the framework. For instance, the test for TensorFlow's `raw_rnn` is only compatible with TensorFlow version 1, yet we are conducting tests in TensorFlow version 2. There were two functions in TensorFlow that fit this case.

For functions that needed to be resampled, we sampled a new function at random from the same frequency decile as the original function.

## 3 Results and discussion

### 3.1 Portability of functions across hardware types

**Overall failure and partial failure rates**: We observe rates of failure for all libraries we benchmark across hardware types. However, the rate of failure differs between frameworks. As seen in Table 1, on GPUs TensorFlow had the highest failure rate with a total of 21.54% complete and partial failures. On TPUs PyTorch has the highest failure rate with a remarkable total of 44.44% complete and partial failures. Across both platforms, we observe the lowest rate of failure for JAX with 1.59% complete failure on the GPU and 3.17% complete failure on the TPU. In particular, PyTorch's TPU failure rates stand out, as double the failure rate of TensorFlow and the highest failure rate overall.

**Failure rate across different percentiles**: One of the questions we wanted to explore was whether portability was impacted by the frequency of use of functions. Our expectation was that the more heavily used a function was, the more portable it would be given the incentive to support the top use cases. However, as shown in Figure 1, there is a fairly consistent failure and partial failure rate across

Table 2: Comparison of the latency in milliseconds for the two functions with the greatest and least increase in latency in TensorFlow, PyTorch, and JAX on GPU and TPU. The table is ordered by the ratio GPU/TPU in descending order, and the top two biggest ratio functions are highlighted. Note that values are rounded to 3 decimal places.

| | Function | GPU | TPU | TPU/GPU |
|---|---|---|---|---|
| Tensorflow | `tf.linalg.svd` | 0.931 | 112.843 | 121.206 |
| | `tf.math.reduce_logsumexp` | 13.028 | 474.586 | 36.428 |
| | `tf.estimator.LoggingTensorHook` | 0.042 | 0.038 | 0.905 |
| | `tf.compat.v1.Session.run` | 5.722 | 3.804 | 0.665 |
| PyTorch | `torch.argsort` | 0.157 | 948.210 | 6039.554 |
| | `torch.optim.Adamax` | 0.069 | 392.712 | 5691.478 |
| | `torch.cuda` | 0.041 | 0.061 | 1.488 |
| | `torch.nn.Conv2d` | 46.053 | 67.081 | 1.457 |
| JAX | `jax.named_call` | 0.007 | 0.012 | 1.714 |
| | `jax.numpy.array` | 0.435 | 0.638 | 1.467 |
| | `jax.numpy.cos` | 172.002 | 26.102 | 0.152 |
| | `jax.numpy.sqrt` | 98.118 | 13.860 | 0.141 |

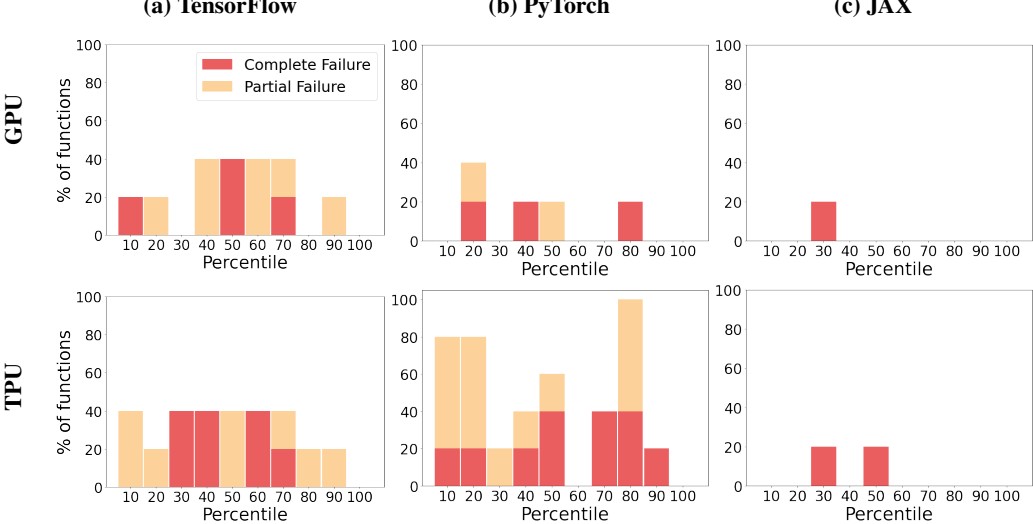

Figure 4: Complete and partial failure rates by percentile bucket for TensorFlow, PyTorch, and JAX functions on GPU and TPU. Note that charts include functions across deciles.

deciles. This holds for all libraries, which suggests that frequency of use has not had a significant effect on the prioritization of support across hardware types.

**Failure rate of top-20 functions**: To further explore whether the frequency of use has influenced portability, we directly compare rates of portability for the top 20 functions vs. other deciles. In Table 4, we observe that some libraries like JAX have 0% failure rates in the top-20 and low overall failure rates across all functions. However, surprisingly on TPUs, PyTorch actually presents slightly higher failure rates in the top 20 functions than across all functions (46% vs 44%). We also observe that on GPUs, TensorFlow also presents a considerably higher rate of failure in the top-20 functions (33% vs 22%). Across the board, we observe the rates of error between the deciles and the top 20 are quite similar showing even the most used functions do not benefit from greatly increased portability.

**Comparing GPUs generations:** When analyzing the portability success and failure rates of Tensor-Flow, PyTorch, and JAX functions across T4 and A100 GPUs, we observe surprisingly similar trends

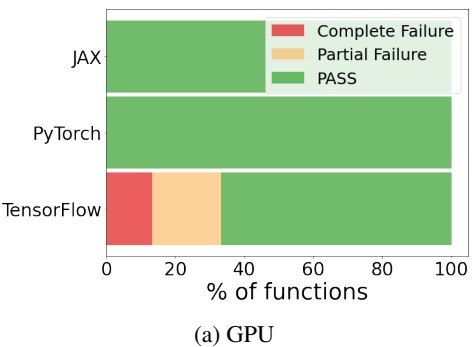 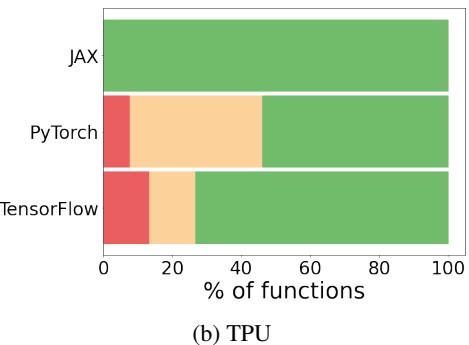

(a) GPU                                           (b) TPU

Figure 5: Complete and partial failure rates for the top 20 functions in TensorFlow, PyTorch, and JAX functions on GPU and TPU.

| Comparison of GPU A100 and T4 Failure and Success Rates | | | | | | |
|---|---|---|---|---|---|---|
| | **T4** | | | **A100** | | |
| | **Success** | **Failure** | | **Success** | **Failure** | |
| | Pass | Partial | Complete | Pass | Partial | Complete |
| TensorFlow | 78% | 8% | 14% | 79% | 9% | 12% |
| PyTorch | 92% | 3% | 5% | 92% | 3% | 5% |
| JAX | 98% | 0% | 2% | 97% | 0% | 3% |

Table 3: Comparison of portability success and failure rates of a random stratified sample of Tensor-Flow, PyTorch, and JAX functions across T4s and A100s.

between the two hardware generations, differing by only up to 1% for TensorFlow and JAX as shown in Table 3. Success rates remain consistently high for all frameworks on both GPUs, indicating robust compatibility. The percentages of Partial and Complete Failures also exhibit comparability across the GPUs and frameworks. This is concerning as it indicates that the advancements in A100 architecture have minimal influence on the overall portability.

**First class citizen effect on different hardware**: One noted effect we see in our results could be described as a *first class citizen effect*. Or simply frameworks built for a device or compilation target perform much better in that environment. The most striking example of this is in JAX on TPUs. As seen in Table 1 in JAX, we see a much lower rate of errors on TPUs when compared to other frameworks with only 3% of functions failing. This is likely due to JAX being built with XLA as a target in mind. We see a similar but less pronounced effect with TensorFlow when compared to PyTorch. TensorFlow was one of the original targets for XLA and thus performed decently well on them with 29% of functions failing when compared to PyTorch which has 44% of functions failing. The first-class citizen effect is less pronounced in TensorFlow, likely due to the age of the framework and the newer JAX giving the teams at Google a chance to rethink what XLA as a compilation target looks like. Compare both of these to PyTorch, and you can see a significant difference. PyTorch is a framework where XLA support was tacked on in a separate library, and it very much shows.

**Reason for failure**: We observe rates of failure for all libraries we benchmark across hardware types. To understand better what impacts hardware portability, we annotate failures into categories. We briefly describe each below:

- **Type failure**: Some types are not implemented in a given device. For instance, in PyTorch, while a TPU tensor might claim to be a double, it will always be represented by a Float instead. Another example of this is in the case of the PyTorch function `SmoothL1Loss`, which on GPUs we attempt to call on the `bFloat16` type. However, this is not implemented and fails.

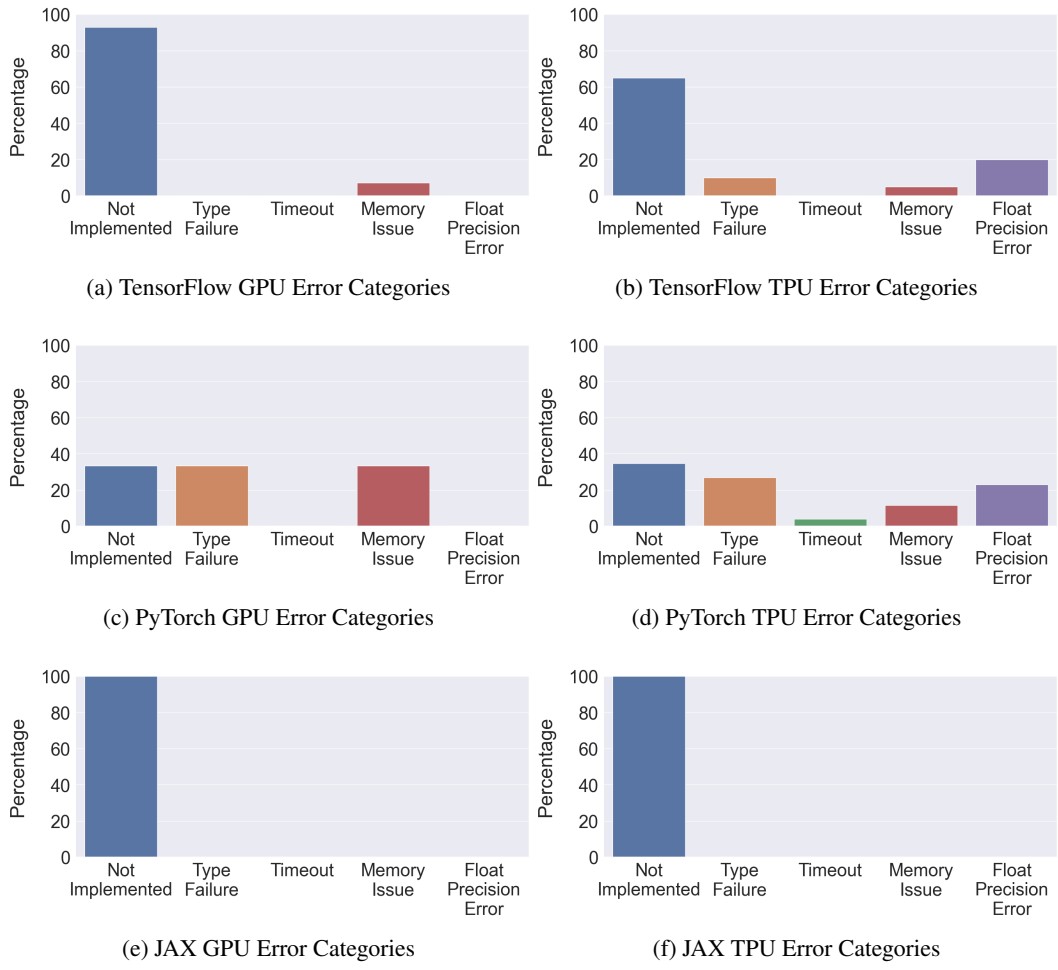

Figure 6: Percentage of failure categories per framework device pair.

- **Not implemented**: Some operations do not have kernels implemented on TPUs or GPUs at all or for specific categories of inputs. For example, on the TensorFlow `numpy_function`, the kernel `PyFuncStateless` is not implemented.

- **Timeout**: We set a threshold of 15 minutes for each test. If a test goes above that, we automatically kill it and record it as a timeout failure. Given the minimal test code and the size of the test inputs (designed to run quickly), we believe 15 minutes was conservatively long.

- **Memory issue**: Captures all cases where memory was attempted to be allocated or accessed as part of the operation and failed. For example, PyTorch `Dataset` attempted to use `pin_memory`, but this does not work on a TPU.

- **Float precision error**: TPUs have a special float class called `bFloat16`, which has fewer mantissa bits and more bits for the exponent. This allows for much smaller and larger values but at the cost of floating point precision. This can break assertions in the tests.

As shown in Figure 6, the most common reason for failure across all frameworks is the `Not Implemented` error, which is pronounced in TensorFlow and JAX, accounting for over 60% of failures. Moreover, PyTorch has a distinctive rise in `Type Failures`, contributing to more than 30% of its failures, a rate noticeably higher than the almost negligible or at most nearly 10% in other frameworks. Both TensorFlow and PyTorch exhibit a relatively low failure rate due to `Memory Issues` and `Type Failures`. As expected, the `Float Precision` error is unique to the TPU, representing around 20% of the failures for both TensorFlow and PyTorch.

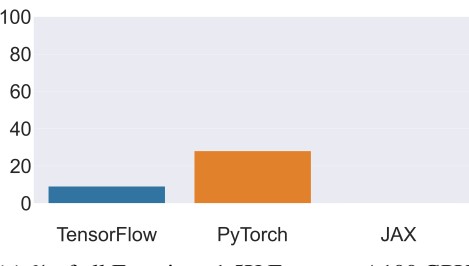
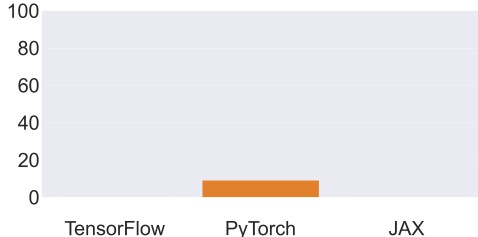

(a) % of all Functions 1.5X Faster on A100 GPU Compared to T4 GPU

(b) % of all Functions 1.5X Faster on v3-8 Compared to v2-8 TPU.

Figure 7: Percentage of functions faster on new GPU/TPU compared with old ones.
.

## 3.2 Efficiency cost to switching tooling

As depicted in Figure 1 and Figure 3, 96% and 100% of TensorFlow and PyTorch functions experience significant deceleration when migrating from GPU to TPU, respectively. Specifically, within TensorFlow and PyTorch, a clear latency gap emerges when functions previously operating on the GPU are transferred to the TPU. As seen in Table 2, the lowest observed latency gap is 0.665 times. This gap tends to be more pronounced for slower operations on the GPU, reaching a maximum latency gap of 121.206 times. In PyTorch, the latency gap is even more prominent, with speed reductions of up to 6039 times when migrating from GPU to TPU. The latency densities also follow this trend, as shown in Figure 2.

In contrast, most functions perform faster on TPU in JAX. When comparing the two devices, there is a minimal latency gap in JAX for both quick and slow operations. The ratio of performance in the migration from GPU to TPU in JAX remains minor, ranging from 0.141 to 1.714 times. In all, we see slowdowns on 100% functions for PyTorch, 96% of functions for TensorFlow, and 8% functions on JAX while moving to the TPU.

There are unique circumstances here that might make this different from using these frameworks in real-life situations (specifically in your standard training situation, you have long-running processes, and in our case, we are running simple functions that finish quickly), but this is clear that the benefits of switching to specialized hardware can be uneven and variable.

## 4 Related work

**Deep learning frameworks**: The rapid adoption and commercial success of Deep learning has spurred the development of software frameworks tailored to deep neural network workloads. Many of the most widely used libraries for machine learning workloads are Python libraries like TensorFlow Abadi et al. [2015], Theano [Team et al., 2016], Chainer [Tokui et al., 2019], MXNet [Chen et al., 2015], PyTorch [Paszke et al., 2019] and JAX [Bradbury et al., 2018]. Despite the variety in frameworks, there has been no study to our knowledge of the difficulty of porting these frameworks between different types of hardware.

**Narrowing of AI research**: The specialization of hardware to create efficiencies for machine learning workloads has created concerns about a narrowing in research contributions. Recent work [Hooker, 2021, Barham and Isard, 2019] suggests that inflexible high-performance kernels and limited programming abstractions are hindering innovative machine learning research. [Hooker, 2021] argues that the availability of accelerator hardware determines the success of ML algorithms potentially more than their intrinsic merits – that the success of ideas hinges on alignment with hardware on software. [Klinger et al., 2022] analyzes arXiv papers and finds that AI research has stagnated in recent years and that AI research involving the private sector tends to be less diverse and more influential than research in academia. Several works [Ahmed and Wahed, 2020] point to the growing compute divide, which impacts accessibility to research and ideas.

**Portability of software frameworks**: Different designs for technology are possible, and some designs are more desirable from an innovation standpoint than others [David et al., 2009]. However, circumstances such as chance events, shortsightedness, and lack of coordination can lead to a

Table 4: Comparison of failure rates between functions in the top 20 and the overall failure rate across all deciles.

|     |               | TensorFlow | PyTorch | JAX |
| --- | ------------- | ---------- | ------- | --- |
| GPU | Top 20        | 33%        | 0%      | 0%  |
|     | All Functions | 22%        | 10%     | 2%  |
| TPU | Top 20        | 27%        | 46%     | 0%  |
|     | All Functions | 30%        | 44%     | 4%  |

situation where an inferior design becomes dominant and difficult to transition away from, even after its limitations become evident [Arthur, 1994, David, 1985]. In the face of uncertainty regarding the advantages and risks associated with different technologies and the desire to avoid getting stuck with an inferior design prematurely, it might be sensible to implement policies that maintain diversity in the technological landscape [David et al., 2009]. A third and final reason to preserve technological mobility and reduce the cost of exploration: innovation involves the creative recombination of ideas, and unusual mixes are often an important source of radical and transformative innovations [Arthur, 2011].

## 5   Limitations

While our work does a great deal to quantify existing gaps in portability, it has some important limitations. Firstly we recorded latency calculations and failure categories on two types of GPUs (A100s and T4s) and two types of TPUs (v2-8 and v3-8). We believe the similar error rates between types of GPUs show that at least for failure rates there is a good deal of consistency between types of GPUs. Worthwhile extensions of this work would include adding more device types to get a more robust view of overall portability and its trend.

Secondly, this paper does not explore in depth why these portability gaps exist. We provide some broad hypotheses on why there might be differences in Section 3.2, but we leave it to future work to pinpoint why these differences exist. One reason for our limitation is due to the lack of access to CUDA internals as it is not completely open source. Understanding the differences in kernels between devices and framework implementations is a daunting task and outside of the scope of this work.

## 6   Conclusion

We benchmark three widely used and adopted machine learning libraries to evaluate the ease of portability across different hardware types. We find large differences in the subset of software operations supported on different types of hardware. We find that PyTorch and TensorFlow, in particular, have pronounced portability issues. On GPUs, 22% of the TensorFlow benchmark functions fail partially or completely. On TPU, a remarkable 44% of PyTorch benchmark functions partially or completely fail. Even where there is portability, significant gaps exist in performance between each framework. We observe that when transferring functions from GPU to TPU, 81.4% of functions in PyTorch exhibit more than 10x slowdown.

Significant work remains to ensure portability and performance between device types. Currently, major slowdowns and broken operations are the norms, and widely used frameworks have overpromised when it comes to portability. This lack of portability has costs to innovation: incentivizing researchers to stick with the tooling they have, which often makes it harder to stray off the beaten path of research ideas. Innovation often occurs when it is cheaper to explore, but our tooling stacks have clearly quantifiable friction that deters exploration. Valuable future work includes developing standardized approaches to machine learning tooling that enable greater portability between different hardware types, and benchmarking additional hardware types. We hope by releasing our benchmark dataset, we can spur greater visibility into what frameworks need more support.

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
