# 7 Appendix

# 8 The Relationship Between Software Portability and Innovation

While some innovation happens de-novo, building something new from scratch, much happens from local adaptation, where an existing innovation is adapted [Eisenhardt and Tabrizi, 1995]. This practice is ubiquitous in machine learning where there is extensive reuse of code and models. Lack of software portability constrains innovation because it means that someone who has previously developed their work in a framework is tied to a particular piece of hardware and may be unable to switch to another advantageous framework if that other framework lacks the functionality/performance needed. While it is hard to count instances of non-invention attributable to hardware because "didn't invent" also means "didn't publish," we can nevertheless see particular examples where the lack of software portability has stifled innovation such as:

1. **Efficiency gains from early exiting** [Teerapittayanon et al., 2017] (Abadi et al. 2016) is a very popular efficiency strategy for avoiding unnecessary computation. However, early exiting has no impact on memory requirements or efficiency when using software stacks that fully instantiate the computation graph prior to running the program (i.e., TensorFlow). Thus this is an optimization that works well in other frameworks but gains us nothing in the case of TensorFlow.

2. **Naive multi-device training distribution strategies** are sensitive to the choice of software stack used. It can have a pronounced impact on differences in dispatch overhead and communication patterns with PyTorch not being able to engage in some distributed workloads [Barham et al., 2022].

3. **Capsule networks** [Sabour et al., 2017] have unique operations like squashing and routing that stray from the matrix multiplies. Capsule networks are far less efficient in TensorFlow, given the requirement for adaptive routing [Barham and Isard, 2019].

4. **Adaptive learning or data pruning**. Both require removing examples from a batch that are estimated not to be important (adaptive pruning does it over the course of training, and data pruning can be a single shot before training). Both techniques have no impact on efficiency when using software stacks that require fixed shapes (i.e., TensorFlow), as instead of changing the batch size on the fly, you need to pad the batch with zeros.

5. **Proximal gradient optimization and variants** [Parikh and Boyd, 2014]. Implementing these techniques in PyTorch is straightforward due to PyTorch's flexible design granting granular control over the training loop. Conversely, Keras abstracts much of the underlying intricacies, which can limit the direct control users have over specific training loop customizations.

We will continue to curate a list of examples where lack of software portability has impacted innovation in research. If you have other examples we should add to this list, please reach out to the authors of this paper.

# 9 Discussion of Differences in Latency

While an in-depth analysis of differences between GPU and TPU kernels of functions is beyond the scope of our paper, we wanted to categorize some high-level reasons for differences as a starting point for discussion. We note these are anecdotal observation, but may be of interest to the reader as a starting point for discussion.

Broadly, we expect slowdowns to be attributed to one of the following categories:

1. **Misalignment between workload and implementation:** frameworks and hardware may assume certain usage patterns (e.g., uniform input sizes) that are mismatched with actual workloads.

2. **Memory architectures:** The substantially different memory architecture choices made by TPU and GPU architects advantage particular data structures and operations, making framework optimizations uneven in their effectiveness [Zhu et al., 2020].

3. **Bottlenecks:** Unimplemented features in some frameworks can lead to data transfer bottlenecks that lead to the full performance not being able to be taken advantage of.

**PyTorch Latency Differences** For TPUs, we observe **long data transfer times** between the TPUs memory and CPUs can be a bottleneck. This is a problem that is much worse in PyTorch than TensorFlow due to a lack of an infeed processing implementation. TensorFlow specifically runs input processing on the CPU while TPU operations take place. PyTorch has chosen not to implement this, which makes TPUs slower than GPUs for PyTorch. Another contributing bottleneck for TPUs is **kernel recompilation on dynamic shapes** which can lead to slower results in tests that use dynamic shapes when running on TPUs.

**TensorFlow Latency Differences kernel recompilation on dynamic shapes** is also a contributing factor for TensorFlow. This leads to our greatest latency difference in TensorFlow on the SVD function. **Data transfer pauses:** While input processing is implemented in TensorFlow, data transfer remains a bottleneck. In some cases, this data preparation and transfer can take longer than the XLA process itself. **Unequal Speedups Due to Specialization:** The largest benefits of TPUs will be on operations involving matrix multiplication. For other operations, large speed-ups are not ensured.

**Performance Comparison Across Hardware Versions:** Referring to Figure 7, 9.09% of TensorFlow functions exhibit a 1.5X performance enhancement when transitioning from a T4 GPU to a A100 GPU. Additionally, 28.07% and 9.09% of PyTorch functions achieve a 1.5X speed improvement when operating newer GPU and TPU versions, respectively. In contrast, JAX functions display minimal gains of just 0.05% on the GPU and 0.02% on the TPU.

## 10 Why Functions

To avoid overfitting to specific machine learning workloads that may not capture future machine learning research directions, we **evaluate the portability of functions and not scripts**. A major concern when overly focusing on popular architectures or tasks is the sidelining the diverse range of code and ideas that researchers are exploring, some of which might not have reached popularity or high optimization levels. In addition, choosing to analyze workloads instead of functions would have posed several challenges for fairly comparing frameworks:

1. **Analysis can be difficult:** For example, if we have input $x$, go through three functions $F$, $G$, and $H$ in the order $F(G(H(x)))$. If the middle function, which is $G$ in this case, fails because it is not portable, we will not be able to test the function $F$.

2. **Different workloads use different framework versions:** If we use a deprecated function, we might face (1).

3. **Privileging common workloads introduces bias:** The function X might work on a common task, but it might not work in a more niche case. Therefore, the function sampling is much more thorough and thus more suitable for extrapolation.

4. **Operations are the building blocks of ML workloads:** The performance and portability of the operations directly impact the workloads that use them.

## 11 Hardware Evaluation and Device Running Procedures

**Types of Hardware Evaluated**: We primarily ran test suites on a T4 GPU and a v3-8 TPU [Jouppi et al., 2017]. For certain analyses, we utilized an A100 GPU and v2-8 TPU, and we specifically indicate such instances in the charts and tables. Unless otherwise indicated, readers should assume the use of a T4 GPU and a v3-8 TPU.

**Ensuring operations executed on correct device**: To ensure that PyTorch, TensorFlow, and JAX tests ran on the right hardware, we provided a device environment variable, which we then referred to in test helpers and startup code to force tests to be on the correct device.

This ensures that operations are not split between multiple devices but instead run on a single device. This was necessary because many tests specifically test transferring values between the CPU and another device, whereas our goal is to establish the viability of running a function on a single device.

We include more details in the appendix Section 13 about the technical implementation of ensuring functions are only run on the device of interest.

**Latency measuring procedure**: For every script and each framework we wrap the relevant operation with `time.perf_counter()`. Before recording the ending time, we include a synchronization point. This will synchronize asynchronous XLA and Cuda operations, allowing the operations to finish before we take the time. We include more details in the appendix in Section 14 about how we implement the synchronization points. We record 3 runs for every test, framework, and device combination. Unless indicated otherwise, results are reported as the average of the 3 runs.

## 12    Data Filtering

We filter the files obtained from the CodeParrot-clean dataset to only include files that import the respective framework using the regexes `'(from.*tensorflow|import.*tensorflow)'`, `'(from.*jax|import.*jax)'`, and `'(from.*torch|import.*torch)'` respectively. These files were subsequently parsed and tokenized [Richter and Wehrheim, 2022] to obtain frequency count for functions. Our goal was to approximate the frequency of functions in everyday engineering usage. While this process was imperfect due to name collisions with identifiers with the same names as framework functions, we managed to get a broad overview of framework function use.

To ensure we capture function calls and variables without including irrelevant pieces of the code, we tokenize each individual Python file. We use code_tokenize [Richter and Wehrheim, 2022] to parse the files and determine if something is an identifier. We run the tokenization on each file and then count the frequencies of relevant identifiers. This ensured we were not looking at import statements, control statements, and other parts of Python code. Instead, we include just function calls, variables, and class usages. Next, we count the frequency of identifiers that were also functions in the respective framework.

While framework functions are our primary interest, classes were necessary to include as well. A frequent pattern is to see classes that act in a very function-like way. A good example of this would be `ReLU` in PyTorch. It is a common pattern in PyTorch to initialize a class and use the resulting instance as a function you can pass input to. With `ReLU`, this would look like this example from the documentation[6]. This brought a certain level of ambiguity because we need to include classes but not all classes are directly relevant. In the interest of ensuring we included everything relevant, we included all class names in the list of all relevant identifiers in a framework.

## 13    Ensuring Functions are Run on Device of Interest

**PyTorch Device Running Procedure**: For PyTorch we leverage the existing `instantiate_device_type_tests` functionality found in the PyTorch test suite. This function allows you to pass a device parameter into each test and use that to set the device of any tensors. We customized this to include XLA TPU support. This involved us overriding the `onlyNativeDeviceTypes` decorator to include the TPU while running. We also created a new decorator to go along with this functionality called `onlyAcceleratedDeviceTypes` for tests that previously had the `onlyCuda` decorator. The `onlyAcceleratedDeviceTypes` decorator ensures that only accelerated devices, such as GPUs and TPUs, are used to run the test. This was necessary because many tests specifically test transferring values between the CPU and another device. So just using `onlyNativeDevices` would not work on those tests.

**TensorFlow Device Running Procedure**: To ensure that all TensorFlow tests were using the correct device we needed to handle the following cases:

1. Tensors running eagerly in the default way.

2. Tensors utilizing graph mode through `self.session` and `self.cached_session` which are built into the TensorFlow test suite.

3. Tensors utilizing `tf.Graph.as_default` to set the graph. This undoes any device setting we do with our contexts and thus needs device setting within `tf.Graph.as_default` call.

---

[6]https://pytorch.org/docs/stable/generated/torch.nn.ReLU.html

To handle these cases, we did the following:

1. Utilize a device context that sets the device to the one specified in our environment variable.

2. Handle tests utilizing `self.session` and `self.cached_session` by monkeypatching the TensorFlow `TestCase` class to use our environment specified device. This was done by overriding the private method `_constrain_devices_and_set_default`.

3. Handle tests utilizing `tf.Graph.as_default` by using a monkeypatched `tf.Graph.as_default` to include the device based upon our environment variable within the call `tf.Graph.as_default`. For most tests this was sufficient. But for several, this broke something else inside these tests. For these specific tests, we created a blacklist upon which we do not apply the monkey patch, and instead, we set the device manually inside the `tf.Graph.as_default` call.

## 14  Measuring Latency on Devices

The synchronization points are as follows:

- TensorFlow on GPU and TPU: We call `.numpy` on the result of the operation which forces all asynchronous operations to finish before times are recorded.

- PyTorch on GPU: We use `torch.cuda.synchronize()` in order to sync the operation.

- PyTorch on TPU: We use `xm.mark_step()` as a synchronization point.

- JAX on GPU and TPU: We use `block_until_ready()` on the output of the operation as a synchronization point.

Table 5: Latency in milliseconds for PyTorch on GPU and TPU. The table is ordered by the ratio GPU/TPU in descending order. Note that values are rounded to 3 decimal places.

|    | Function | GPU | TPU | TPU/GPU |
|----|----------|-----|-----|---------|
| 1  | `torch.argsort` | 0.157 | 948.210 | 6039.554 |
| 2  | `torch.optim.Adamax` | 0.069 | 392.712 | 5691.478 |
| 3  | `torch.fliplr` | 0.201 | 725.480 | 3609.353 |
| 4  | `torch.broadcast_tensors` | 0.044 | 39.030 | 887.045 |
| 5  | `torch.nn.AdaptiveAvgPool3d` | 0.074 | 65.219 | 881.338 |
| 6  | `torch.addr` | 0.106 | 83.030 | 783.302 |
| 7  | `torch.cat` | 0.100 | 64.652 | 646.520 |
| 8  | `torch.optim.LBFGS` | 0.097 | 50.358 | 519.155 |
| 9  | `torch.triangular_solve` | 0.091 | 33.536 | 368.527 |
| 10 | `torch.nn.Module.state_dict` | 0.053 | 19.508 | 368.075 |
| 11 | `torch.nn.Module.zero_grad` | 0.057 | 17.572 | 308.281 |
| 12 | `torch.sum` | 0.084 | 23.921 | 284.774 |
| 13 | `torch.Tensor.is_same_size` | 0.025 | 6.973 | 278.920 |
| 14 | `torch.nn.KLDivLoss` | 0.187 | 48.865 | 261.310 |
| 15 | `torch.nn.LSTMCell` | 0.372 | 93.617 | 251.659 |
| 16 | `torch.moveaxis` | 0.034 | 7.878 | 231.706 |
| 17 | `torch.nn.functional.dropout` | 0.041 | 8.847 | 215.780 |
| 18 | `torch.lt` | 0.055 | 8.857 | 161.036 |
| 19 | `torch.autograd.Variable` | 0.051 | 7.519 | 147.431 |
| 20 | `torch.utils.data.Subset` | 0.069 | 8.722 | 126.406 |
| 21 | `torch.nn.Sequential` | 0.115 | 11.639 | 101.209 |
| 22 | `torch.multinomial` | 1.193 | 115.240 | 96.597 |
| 23 | `torch.nn.Linear` | 0.459 | 37.842 | 82.444 |
| 24 | `torch.nn.BCEWithLogitsLoss` | 0.855 | 55.768 | 65.226 |
| 25 | `torch.diag` | 0.494 | 27.232 | 55.126 |
| 26 | `torch.von_mises.VonMises` | 0.680 | 31.054 | 45.668 |

Continued on next page

| | Function | GPU | TPU | TPU/GPU |
|---|---|---|---|---|
| 27 | `torch.zeros` | 0.252 | 10.577 | 41.972 |
| 28 | `torch.round` | 0.301 | 12.178 | 40.458 |
| 29 | `torch.range` | 0.193 | 5.627 | 29.155 |
| 30 | `torch.autograd.functional.jvp` | 0.788 | 22.298 | 28.297 |
| 31 | `torch.linalg.matrix_rank` | 2.568 | 68.511 | 26.679 |
| 32 | `torch.nn.GELU` | 0.787 | 19.317 | 24.545 |
| 33 | `torch.Tensor.to` | 0.105 | 1.728 | 16.457 |
| 34 | `torch.nn.Transformer` | 216.593 | 3066.136 | 14.156 |
| 35 | `torch.bitwise_not` | 1.925 | 23.535 | 12.226 |
| 36 | `torch.nn.Conv3d` | 0.240 | 1.202 | 5.008 |
| 37 | `torch.slogdet` | 30.606 | 151.092 | 4.937 |
| 38 | `torch.optim.lr_scheduler.ExponentialLR` | 0.037 | 0.157 | 4.243 |
| 39 | `torch.utils.data.Dataset` | 0.035 | 0.117 | 3.343 |
| 40 | `torch.utils.data.ConcatDataset` | 0.031 | 0.092 | 2.968 |
| 41 | `torch.nn.Parameter.register_parameter` | 0.039 | 0.091 | 2.333 |
| 42 | `torch.cuda` | 0.041 | 0.061 | 1.488 |
| 43 | `torch.nn.Conv2d` | 46.053 | 67.081 | 1.457 |

Table 6: Latency in milliseconds for TensorFlow on GPU and TPU. The table is ordered by the ratio GPU/TPU in descending order. Note that values are rounded to 3 decimal places.

| | Function | GPU | TPU | TPU/GPU |
|---|---|---|---|---|
| 1 | `tf.linalg.svd` | 0.931 | 112.843 | 121.206 |
| 2 | `tf.math.reduce_logsumexp` | 13.028 | 474.586 | 36.428 |
| 3 | `tf.nn.conv3d` | 3.596 | 49.867 | 13.867 |
| 4 | `tf.tensor_scatter_nd_update` | 1.625 | 21.626 | 13.308 |
| 5 | `tf.signal.idct` | 6.965 | 87.764 | 12.601 |
| 6 | `tf.python.ops.numpy_ops.clip` | 1.223 | 15.409 | 12.599 |
| 7 | `tf.image.adjust_brightness` | 10.264 | 129.021 | 12.570 |
| 8 | `tf.train.list_variables` | 0.358 | 4.032 | 11.263 |
| 9 | `tf.reshape` | 1.161 | 10.185 | 8.773 |
| 10 | `tf.cast` | 1.214 | 10.546 | 8.687 |
| 11 | `tf.lookup.KeyValueTensorInitializer` | 2.534 | 17.919 | 7.071 |
| 12 | `tf.Tensor.eval` | 3.332 | 22.366 | 6.712 |
| 13 | `tf.range` | 2.108 | 13.782 | 6.538 |
| 14 | `tf.convert_to_tensor` | 2.027 | 9.683 | 4.777 |
| 15 | `tf.sequence_mask` | 10.962 | 46.188 | 4.213 |
| 16 | `tf.compat.v1.distributions.Normal` | 1.871 | 5.340 | 2.854 |
| 17 | `tf.debugging.assert_less` | 5.541 | 14.911 | 2.691 |
| 18 | `tf.math.reduce_mean` | 7.629 | 19.746 | 2.588 |
| 19 | `tf.python.framework.smart_cond` | 3.067 | 7.838 | 2.556 |
| 20 | `tf.compat.v1.test.compute_gradient_error` | 164.855 | 377.978 | 2.293 |
| 21 | `tf.nn.conv2d_transpose` | 72.639 | 152.463 | 2.099 |
| 22 | `tf.dtypes.as_dtype` | 0.009 | 0.018 | 2.000 |
| 23 | `tf.compat.v1.distributions.Normal.survival_function` | 9.521 | 18.128 | 1.904 |
| 24 | `tf.compat.v1.distributions.Normal.param_shapes` | 0.542 | 1.022 | 1.886 |
| 25 | `tf.contrib.framework.nest.map_structure_up_to` | 0.404 | 0.760 | 1.881 |
| 26 | `tf.numpy_function` | 1.572 | 2.954 | 1.879 |
| 27 | `tf.sets.intersection` | 1.926 | 3.606 | 1.872 |
| 28 | `tf.compat.v1.saved_model.simple_save` | 27.784 | 50.100 | 1.803 |
| 29 | `tf.compat.v1.placeholder` | 0.721 | 1.293 | 1.793 |
| 30 | `tf.keras.optimizers.experimental.Adadelta` | 7.951 | 13.918 | 1.750 |
| 31 | `tf.linalg.set_diag` | 1.409 | 2.344 | 1.664 |
| 32 | `tf.compat.v1.variable_scope` | 3.017 | 4.834 | 1.602 |

| | Function | GPU | TPU | TPU/GPU |
|---|---|---|---|---|
| 33 | `tf.constant` | 1.094 | 1.717 | 1.569 |
| 34 | `tf.nn.space_to_batch` | 1.597 | 2.492 | 1.560 |
| 35 | `tf.summary.flush` | 0.823 | 1.274 | 1.548 |
| 36 | `tf.Variable` | 12.660 | 18.490 | 1.461 |
| 37 | `tf.compat.v1.metrics.accuracy` | 26.136 | 38.095 | 1.458 |
| 38 | `tf.compat.v1.get_collection` | 0.015 | 0.021 | 1.400 |
| 39 | `tf.math.igammac` | 0.918 | 1.274 | 1.388 |
| 40 | `tf.test.TestCase.assert_equal` | 0.040 | 0.052 | 1.300 |
| 41 | `tf.compat.v1.global_variables_initializer` | 0.585 | 0.760 | 1.299 |
| 42 | `tf.Graph.as_default` | 0.117 | 0.152 | 1.299 |
| 43 | `tf.train.ExponentialMovingAverage` | 0.021 | 0.025 | 1.190 |
| 44 | `tf.compat.v1.TextLineReader.restore_state` | 1.201 | 1.416 | 1.179 |
| 45 | `tf.distribute.Strategy.get_per_replica_batch_size` | 0.016 | 0.018 | 1.125 |
| 46 | `tf.estimator.CheckpointSaverHook` | 0.051 | 0.055 | 1.078 |
| 47 | `tf.Tensor.get_shape` | 0.040 | 0.042 | 1.050 |
| 48 | `tf.nest.map_structure` | 0.084 | 0.087 | 1.036 |
| 49 | `tf.compat.v1.train.get_global_step` | 0.069 | 0.071 | 1.029 |
| 50 | `tf.estimator.LoggingTensorHook` | 0.042 | 0.038 | 0.905 |
| 51 | `tf.compat.v1.Session.run` | 5.722 | 3.804 | 0.665 |

Table 7: Latency in milliseconds for JAX on GPU and TPU. The table is ordered by the ratio GPU/TPU in descending order. Note that values are rounded to 3 decimal places.

| | Function | GPU | TPU | TPU/GPU |
|---|---|---|---|---|
| 1 | `jax.named_call` | 0.007 | 0.012 | 1.714 |
| 2 | `jax.numpy.array` | 0.435 | 0.638 | 1.467 |
| 3 | `jax.numpy.zeros` | 0.673 | 0.890 | 1.322 |
| 4 | `jax.lax.select` | 197.595 | 225.906 | 1.143 |
| 5 | `jax._src.interpreters.partial_eval` | 0.012 | 0.013 | 1.083 |
| 6 | `jax.core.eval_context()` | 0.005 | 0.005 | 1.000 |
| 7 | `jax.lax.all_gather` | 0.348 | 0.342 | 0.983 |
| 8 | `jax.lax.integer_pow` | 0.197 | 0.190 | 0.964 |
| 9 | `jax.numpy.size` | 0.015 | 0.014 | 0.933 |
| 10 | `jax.tree_util.Partial` | 0.013 | 0.012 | 0.923 |
| 11 | `jax.make_jaxpr` | 2.608 | 2.395 | 0.918 |
| 12 | `jax.numpy.log` | 0.309 | 0.283 | 0.916 |
| 13 | `jax.numpy.isscalar` | 0.010 | 0.009 | 0.900 |
| 14 | `jax.tree_util.tree_unflatten` | 0.008 | 0.007 | 0.875 |
| 15 | `jax.vjp` | 9.834 | 8.565 | 0.871 |
| 16 | `jax.numpy.einsum_path` | 0.282 | 0.243 | 0.862 |
| 17 | `jax.numpy.delete` | 0.013 | 0.011 | 0.846 |
| 18 | `jax._src.interpreters.partial_eval.trace_to_jax_pr_dynamic` | 0.380 | 0.308 | 0.811 |
| 19 | `jax.scipy.stats.norm.cdf` | 0.005 | 0.004 | 0.800 |
| 20 | `jax.lax.stop_gradient` | 0.166 | 0.132 | 0.795 |
| 21 | `jax.numpy.reshape` | 3.714 | 2.845 | 0.766 |
| 22 | `jax.numpy.average` | 0.004 | 0.003 | 0.750 |
| 23 | `jax.disable_jit` | 0.027 | 0.020 | 0.741 |
| 24 | `jax.tree_util.tree_map` | 0.051 | 0.033 | 0.647 |
| 25 | `jax._src.core.get_aval` | 0.073 | 0.047 | 0.644 |
| 26 | `jax.scipy.signal.convolve2d` | 401.254 | 206.587 | 0.515 |
| 27 | `jax.lax.erf` | 14.456 | 7.296 | 0.505 |
| 28 | `jax.scipy.special.ndtr` | 77.206 | 33.315 | 0.432 |
| 29 | `jax.numpy.convolve` | 211.884 | 86.244 | 0.407 |
| 30 | `jax.numpy.linalg.svd` | 794.990 | 301.459 | 0.379 |

| | Function | GPU | TPU | TPU/GPU |
|---|---|---|---|---|
| 31 | `jax.numpy.compress` | 133.591 | 45.108 | 0.338 |
| 32 | `jax.numpy.stack` | 145.019 | 44.748 | 0.309 |
| 33 | `jax.scipy.special.i0` | 389.510 | 118.412 | 0.304 |
| 34 | `jax.numpy.var` | 309.684 | 93.786 | 0.303 |
| 35 | `jax.numpy.tril` | 192.110 | 57.478 | 0.299 |
| 36 | `jax.numpy.sum` | 63.060 | 18.442 | 0.292 |
| 37 | `jax.numpy.triu_indices` | 357.124 | 103.997 | 0.291 |
| 38 | `jax.numpy.power` | 114.465 | 33.004 | 0.288 |
| 39 | `jax.numpy.ones` | 42.366 | 12.144 | 0.287 |
| 40 | `jax.lax.pmax` | 69.271 | 19.671 | 0.284 |
| 41 | `jax.numpy.max` | 147.327 | 39.700 | 0.269 |
| 42 | `jax.scipy.linalg.lu` | 654.291 | 164.111 | 0.251 |
| 43 | `jax.numpy.prod` | 180.961 | 45.276 | 0.250 |
| 44 | `jax.lax.slice_in_dim` | 30.590 | 7.504 | 0.245 |
| 45 | `jax.lax.bitwise_and` | 109.078 | 25.885 | 0.237 |
| 46 | `jax.numpy.tril_indices_from` | 923.872 | 217.183 | 0.235 |
| 47 | `jax.numpy.arange` | 466.126 | 107.242 | 0.230 |
| 48 | `jax.numpy.add` | 130.517 | 29.742 | 0.228 |
| 49 | `jax.numpy.all` | 206.400 | 46.815 | 0.227 |
| 50 | `jax.scipy.special.gammaln` | 161.529 | 34.914 | 0.216 |
| 51 | `jax.numpy.mean` | 222.164 | 47.596 | 0.214 |
| 52 | `jax.numpy.flip` | 75.649 | 14.542 | 0.192 |
| 53 | `jax.numpy.split` | 252.392 | 46.583 | 0.185 |
| 54 | `jax.numpy.fliplr` | 64.010 | 11.766 | 0.184 |
| 55 | `jax.lax.top_k` | 122.615 | 22.130 | 0.180 |
| 56 | `jax.numpy.exp` | 45.239 | 7.792 | 0.172 |
| 57 | `jax.lax.ge` | 90.087 | 15.302 | 0.170 |
| 58 | `jax.nn.one_hot` | 138.935 | 23.335 | 0.168 |
| 59 | `jax.random.PRNGKey` | 1485.077 | 227.995 | 0.154 |
| 60 | `jax.numpy.cos` | 172.002 | 26.102 | 0.152 |
| 61 | `jax.numpy.sqrt` | 98.118 | 13.860 | 0.141 |

Table 8: Comparison of TPUs and GPUs in terms of failure and success counts.

| | **Comparison of TPU and GPU in terms of failure and success counts** | | | | | |
|---|---|---|---|---|---|---|
| | **GPUs** | | | **TPUs** | | |
| | Partial Failure | Complete Failure | Success | Partial Failure | Complete Failure | Success |
| TensorFlow | 5/65 | 9/65 | 51/65 | 10/65 | 9/65 | 46/65 |
| PyTorch | 2/63 | 3/63 | 58/63 | 17/63 | 11/63 | 36/63 |
| JAX | 0/63 | 1/63 | 62/63 | 0/63 | 2/63 | 61/63 |

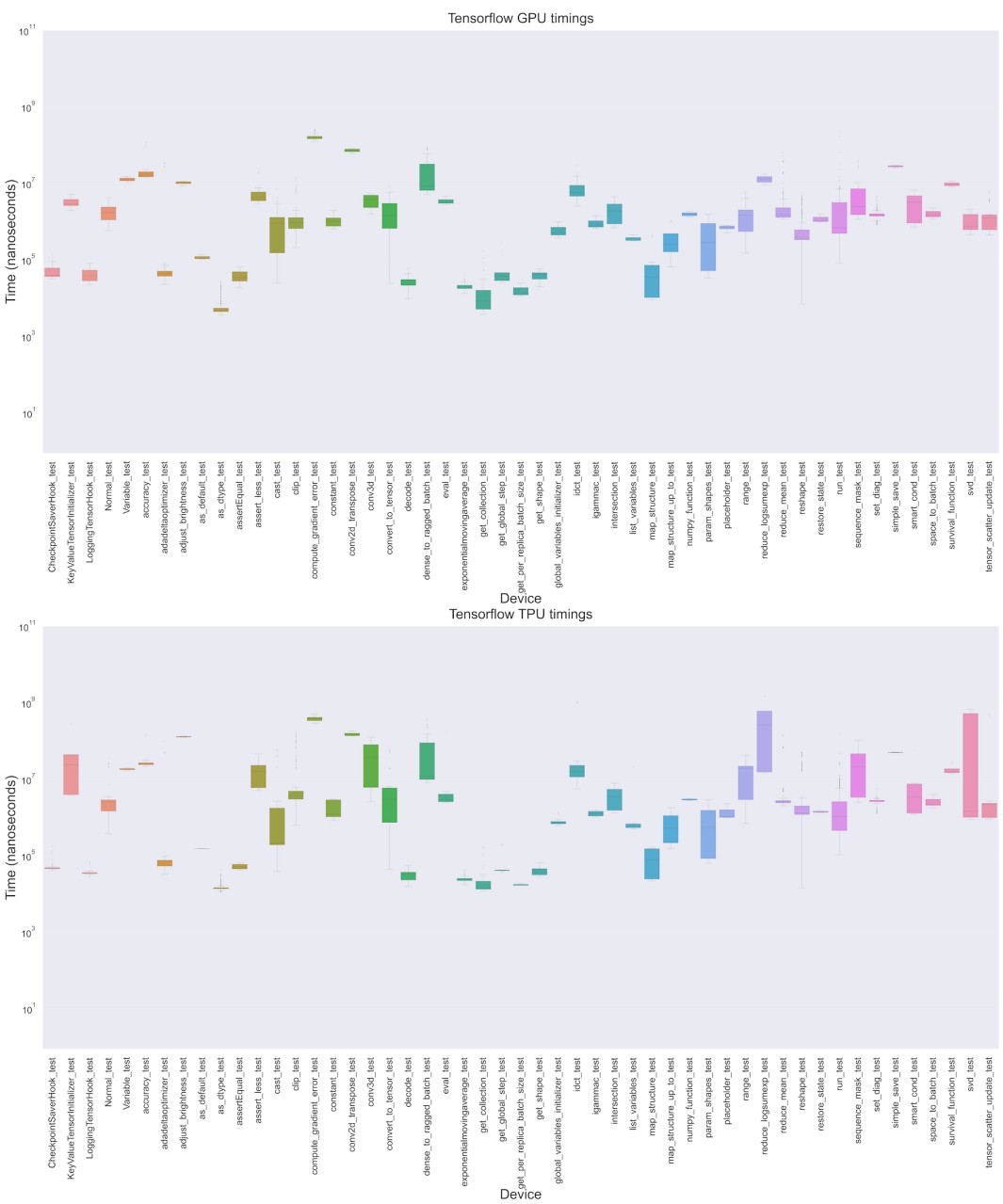

Figure 8: Distribution of times for operations in TensorFlow on GPUs and TPUs.

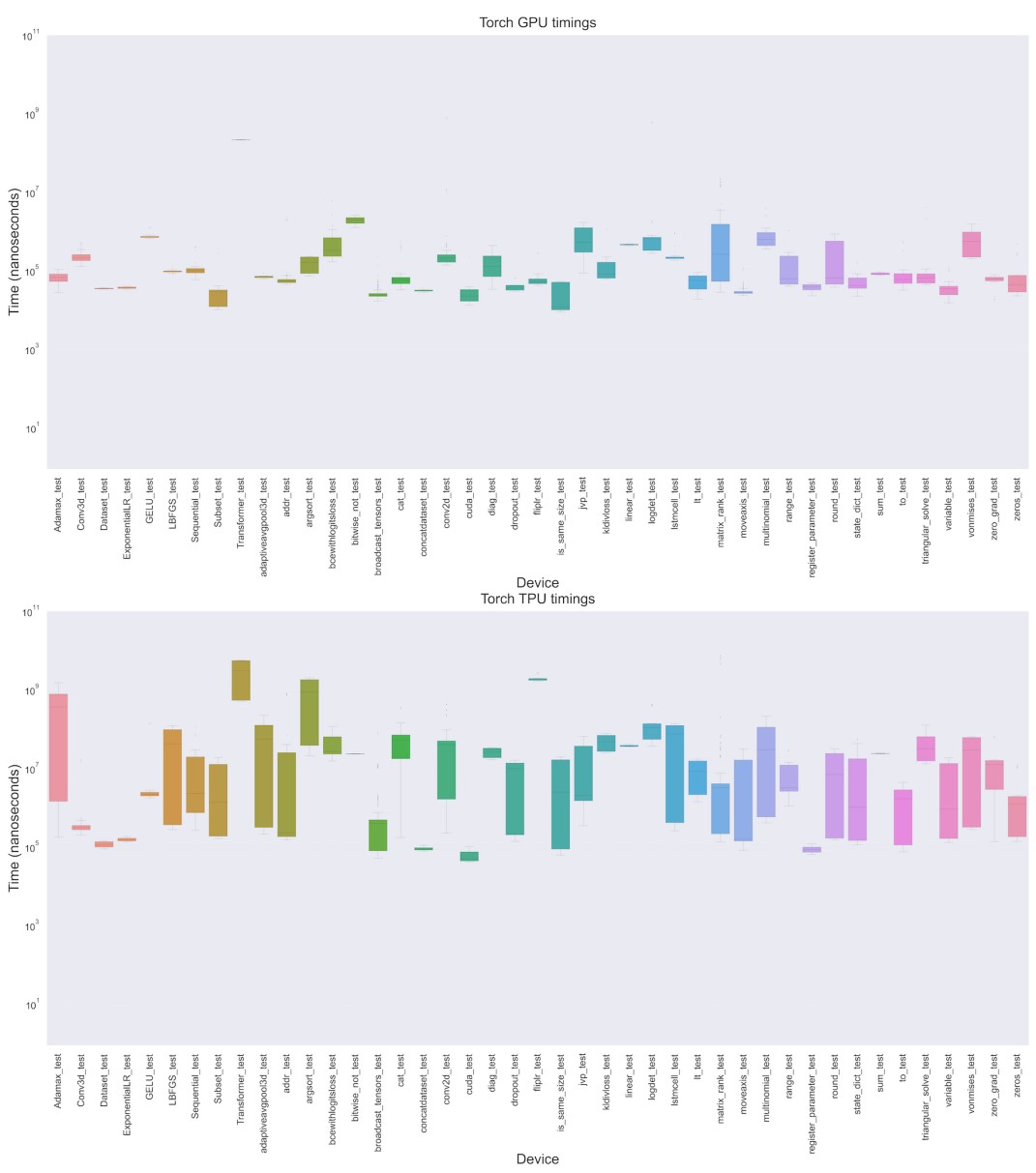

Figure 9: Distribution of times for operations in PyTorch on GPUs and TPUs.

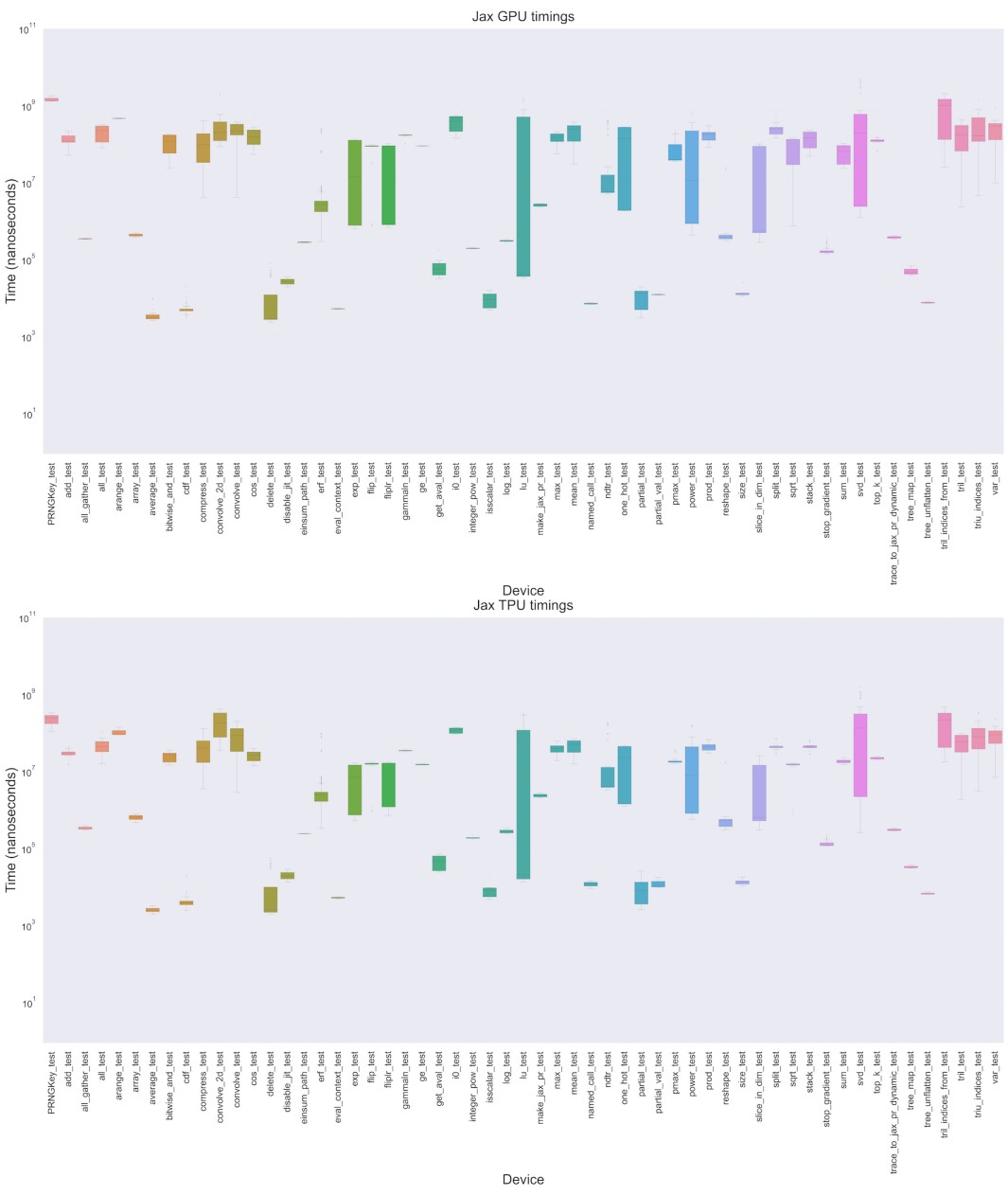

Figure 10: Distribution of times for operations in JAX on GPUs and TPUs.