# OpenReview forum: "The Grand Illusion: The Myth of Software Portability and Implications for ML Progress."
_NeurIPS.cc/2023/Conference — NeurIPS 2023 poster_

### Official Review · Reviewer_fKz5 · 2023-07-06

**Soundness:** 2 fair
**Presentation:** 2 fair
**Contribution:** 1 poor
**Rating:** 4
**Confidence:** 4

**Summary:**

The paper explores the combination between hardware and software for machine learning, and ask the question:  How portable are popular ML software frameworks?  The authors conduct a quantity study of the portability of mainstream ML frameworks across different hardware types. With the experiment result, they claim that machine-learning frameworks can lose more than 40% of their key functions when ported to other hardware, and significantly slowdown if portable.  Collectively, the results reveal how costly straying from a narrow set of hardware-software combinations can be and suggest that specialization of hardware impedes innovation in machine learning research.

**Strengths:**

1. The paper raise a concern for lack of machine learning portable quantity, which is not payed enough attention to in the current research.
2. This paper reports large-scale experimental results to quantity the hardware-software combination, which can be reference for the industry developers.

**Weaknesses:**

1. The lack of portability with different hardware/software structures is widely known knowledge. For example, APPs work on Arm/Android systems is difficult to work on the X86/Windows system. Under mature commercial circumstances, the paper lacks proving why portable is important and necessary in ML.
2. The paper concludes, “Specialization of hardware impedes innovation in machine learning research”, which is not been proved in this paper. According to the No Free Lunch theory in computer architecture studies, domain-specific hardware needs to find a tradeoff between generality and efficiency on specific workloads. It is not realistic to design hardware supporting every workload as well as other hardware frameworks, while still keeping its own advantage in some domains.
3. The paper needs to focus more on the quantity standard for ML portable. The method now only considers the slowdowns of the software framework APIs, but not the actual tasks or NNs (such as Resnet or VGG). The standard is confusing and the authors need more analysis or evidence.
4. The paper is not easy to follow. It is not clear to understand the significance and originality of the contribution.

**Questions:**

1. Paper significance: How the lack of portability of software frameworks is significant in ML? It is not easy to understand since there are no realistic evidence. What does the quantity result suggest comparing to the intuitive result? We would like to know more about why quantity is important here.
2. Methods: Why the paper uses the framework operations instead of NN workloads for evaluation? The codes need to be manually check to port on different hardware.

**Limitations:**

The authors have not discussed their limitations.

---

> ### Author Rebuttal · Authors · 2023-08-10
>
> # Rebuttal (fKz5):
>
> We thank the reviewer for their feedback, including their positive observation that this work "raise[d] a concern for lack of machine learning portable quantity, which is not payed enough attention to in the current research" and the positive view of the experimental rigor and breadth of our work. These experimental results were the core of our research. We believe we are among the first to quantify the lack of portability between machine learning frameworks. Moreover, we will release the dataset to the ML community.
>
> From reviewer fKz5, we are rebutting the following claims:
>
> 1. **"The lack of portability with different hardware/software structures is widely known knowledge."**: While knowledge of hardware disparities may be known by some, we believe the magnitude of the findings presented in this paper are not known. We believe our findings are striking: PyTorch and TensorFlow have portability issues across GPUs and TPUs, with performance discrepancies over 10x slowdown when moved from GPU to TPU for PyTorch; meanwhile, JAX functions, designed with TPUs in mind, run 91.8% faster on TPUs. Our contribution is to offer a rigorous evaluation and benchmark this issue, which is in keeping with the goals of the Evaluation track.
> 2. **"Under mature commercial circumstances, the paper lacks proving why portable is important and necessary in ML."**: Our primary focus is on the portability of ML frameworks for researchers, as stated in Lines 29-30. We are interested in how a lack of portability can increase the barrier of entry for new ideas. A lack of software portability means that researchers are locked into certain hardware/software combinations. This can bias against certain ideas and advantage others. This has been widely discussed in research, including Hooker et al. (2021) and Barham et al. (2019).
>
>    The effect on developers and researchers of significant differences in hardware latency is that ideas overfit certain tooling stacks and fail to transfer to other software stacks. This can turn the act of writing machine learning code into an arcane art of understanding framework sharp edges and hardware complexity.
>
> 3. **"The method now only considers the slowdowns of the software framework APIs, but not the actual tasks or NNs (such as Resnet or VGG)"**: Our objective was to develop an analysis that would broadly cover a wide variety of workloads. A major concern when overly focusing on popular architectures or tasks is that we might sideline the diverse range of code and ideas that researchers are exploring, some of which might not have reached popularity or high optimization levels. To address our concerns about workload representation, we sampled our functions from a distribution based on function frequency. This was designed not to underestimate workloads, thus ensuring a balanced view of the performance landscape across the range of ideas researchers may want to try.
>
>    Analyzing workloads instead of individual functions also poses additional challenges in terms of the rigor of our analysis. We will add the following reasoning to the final manuscript to make more clear:
>
>    1. Sometimes, it is not possible to analyze. For example, if we have input x, go through three functions F, G, and H in the order F(G(H(x))). If the middle function, which is G in this case, fails because it is not portable, we will not be able to test the function F.
>    2. Different workloads also use different framework versions. Therefore, if we use a deprecated function, we might face (1).
>    3. It is biased. The function X might work on the common task, but it might not work in the niche case. Therefore, the test case itself is much more thorough and thus more suitable for testing.
>    4. Operations are the building blocks of ML workloads. The performance and portability of the operations directly impact the workloads that use them.
>
> 4. **"Domain-specific hardware needs to find a tradeoff between generality and efficiency on specific workloads. It is not realistic to design hardware supporting every workload as well as other hardware frameworks, while still keeping its own advantage in some domains"**: While it does not make sense to implement ML operations for every feasible piece of hardware, it is not an unrealistic observation to point out that the current state of ML frameworks can increase friction in the adoption and exploration of ideas across hardware because of the severe drops in performance.
> 5. **"How the lack of portability of software frameworks is significant in ML? It is not easy to understand since there are no realistic evidence. What does the quantity result suggest comparing to the intuitive result? We would like to know more about why quantity is important here."**: We motivate our work by citing prior works (Hooker et al. (2021) and P. Barham et al. (2019)), which make a strong case for why the lack of portability is significant. As for the quantities themselves, we believe quantifying latency differences over multiple runs tells a compelling story about the portability differences between frameworks.
> 6. **"The authors have not discussed their limitations."**: We thank the reviewer for pointing this out! Our research has three limitations. Firstly, we used only one type of GPU and TPU. During the rebuttal, we added an additional variant, which we believe mitigates some of the limitations of the original submission. Secondly, our current focus is on portability for researchers, so a future direction would be to focus on commercial/application-based portability, which would include analysis involving CPUs. Lastly, we couldn't discern the underlying reasons for the observed performance mitigation when transferring functions across devices since we don't have access to the source code (CUDA is not open source). Exploring this aspect would be valuable for the ML community. We will add these limitations in our final manuscript.

---

> > ### Author Response · Authors · 2023-08-16
> >
> > Now that discussion is underway, we wanted to ask fKz5 if there are any follow-up points you would like further clarity on. During the rebuttal period, we carried out additional experiments and are here to ensure that we have addressed your questions or doubts regarding the limitation of our methodology, paper significance, the tradeoff between generality and efficiency, portability under commercial circumstances, and the quantity standard for ML portability. If everything is clear and the recent experiments have addressed your concerns, we kindly ask fKz5 to consider increasing their score to reflect these changes.

---

> > ### Comment · Reviewer_fKz5 · 2023-08-18
> > **Thanks for the response!**
> >
> > Sorry for my late response. I thank the authors for their detailed rebuttal. My major concern remains in the novelty and significance of this work.
> >
> > 1. The fact “lack of portability between machine learning frameworks” is not a novel claim. In this paper, the experiment gives more quantity evidence to the fact, and it does not contribute much to the research community. The key question  to the authors should be "Why the portability gap should be evaluated in quantity since it is a known knowledge?" The experiment results should be further analyzed. For example, researchers only know the lack of portability exists, but "Why" and "How" the lack of portability occurs should be studied.
> >
> > 2. This paper claims “how a lack of portability (between hardware) can increase the barrier of entry for new ideas”, but it is not been proved in the paper. How well functions (and more broadly models) map from one framework to another seems to be more important to researchers with new ideas. For example, the quantity result cannot prove what specific kind of new ideas are constrained when researchers are locked into certain hardware/software combinations. The paper cited some discussions in the paper, but the quantity results in this paper do not contribute much to the claim over the related discussions.
> >
> > 3. The fact "TPUs and GPUs support some software APIs better than others" is not a novel claim. This paper quantifies the gap between the APIs, but it is not clear what to prove. According to Amdahl’s Law, partial results (such as APIs) do not tell the exact story of real-world programs. The quantity results do not yet support the claim that there are severe drops in performance for researchers and thus increase the barrier of entry for new ideas.
> >
> > 4. Overall, the paper contributes to quantifying the lack of hardware portability, but my major concern is why the quantity results are significant to the research community.

---

> > > ### Author Response · Authors · 2023-08-19
> > >
> > > Thank you to **fKz5** for the response and the additional detail about what would convince them about the paper’s significance and novelty. In our response, we’d like to specifically address several of the issues raised: **(i)** whether it constrains innovation, **(ii)** whether this phenomenon is well-known, and **(iii)** whether quantifying this effect (rather than just noting its existence) is important for broader decisions.
> > >
> > > **(i) Does a lack of portability constrain innovation?**
> > > The reviewer is concerned that our paper does not show evidence of the constraining of innovation. While not explicit, we would argue that we are doing this implicitly, and we provide some examples here to justify why these implicit limitations would be consequential.
> > >
> > > While some innovation happens de-novo, building something new from scratch, much happens from local adaptation, where an existing innovation is adapted (Eisenhardt and Behnam N. Tabrizi 1995). Such practice is, of course, very common in ML where there is extensive reuse of code, models, etc. Our paper directly implies a constraining of innovation because it means that someone who has previously developed their work in a framework, like TensorFlow, tied to a particular piece of hardware may be unable to switch to another advantageous framework if that other framework lacks functionality/performance needed or will overfit their ideas to their tooling stack they have.
> > >
> > > While it is hard to directly count instances of non-invention because “didn’t invent” also means “didn’t publish,” we can nevertheless see particular examples where the lack of software portability has stifled innovation:
> > > * **Early exiting (Abadi et al. 2016, Teerapittayanon et al. 2017)** is a very popular efficiency strategy for avoiding unnecessary computation. But early exiting has no impact on memory requirements or efficiency when using software stacks that fully instantiate the computation graph prior to running the program (i.e., TensorFlow). Thus this is an optimization that works well in other frameworks but gains us nothing in the case of TensorFlow.
> > > * **Naive multi-device training distribution strategies** are sensitive to the choice of software stack used. It can have a pronounced impact on differences in dispatch overhead and communication patterns with Pytorch not being able to engage in some distributed workloads (Barham et al., 2022).
> > > * **Capsule networks (Sabour et al. 2017)** have unique operations like squashing and routing that stray from the matrix multiplies. Capsule networks are far less efficient in TensorFlow, given the requirement for adaptive routing.
> > > * **Adaptive learning or data pruning**. Both require removing examples from a batch that are estimated not to be important (adaptive pruning does it over the course of training, and data pruning can be a single shot before training). Both techniques have no impact on efficiency when using software stacks that require fixed shapes (i.e., TensorFlow), as instead of changing the batch size on the fly, you need to pad the batch with zeros.
> > > * **Proximal gradient optimization and variants (Parikh and Boyd 2014)**. Implementing these techniques in PyTorch is straightforward due to Pytorch’s flexible design granting granular control over the training loop. Conversely, Keras abstracts much of the underlying intricacies, which can limit the direct control users have over specific training loop customizations.
> > >
> > > All these examples are promising and important research directions that are impacted by the lack of portability in tooling. In the next response to **fKz5**, we will respond to the remaining two concerns of **(ii)** whether this phenomenon is well-known, and **(iii)** whether quantifying this effect (rather than just noting its existence) is important for broader decisions.

---

> > > > ### Author Response · Authors · 2023-08-19
> > > >
> > > > In this part of the response, we will address two of the issues raised by **fKz5: (ii)** whether this phenomenon is well-known, and **(iii)** whether quantifying this effect (rather than just noting its existence) is important for broader decisions.
> > > >
> > > > **(ii) Is the lack of portability phenomenon well-known?**
> > > >
> > > > While we agree that there is some knowledge about the lack of portability, we would argue that this knowledge is neither widely spread nor understood with sufficient detail to allow informed decision-making.
> > > >
> > > > A simple search of common problem posting boards shows an abundance of examples of ML researchers failing to realize that their attempt to innovate a system has been stymied by portability constraints. While such examples are only anecdotal, they point to a real challenge: how does one decide which framework is right? We would argue that it is nearly impossible to make such choices confidently without the type of evaluation we present in this paper. One would need to know how portable the particular functions that they want to use are, and how much performance penalty they might pay because of their framework choice.
> > > >
> > > > Precisely because there has been a lack of quantitative evaluation, users of these frameworks lack the tools needed to make the best implementation choices. The importance of our work lies in quantifying and evaluating the lack of portability as highlighted by the goals of the track this work is in: **Evaluation (methodology, meta studies, replicability and validity)**.
> > > >
> > > > **(iii) Is quantifying this effect important for big decisions (beyond simply noting its existence)**
> > > >
> > > > Here we present two arguments for why the quantified evaluation of portability is important for larger strategic decisions facing the broader ML community.
> > > >
> > > > As a broader community, it is important to understand not only which ML users would benefit from using one framework over another, but to understand when new frameworks are needed. But building such a framework is an enormous investment, so it should only be undertaken if the need is important. Similarly, users will have to pay a switching cost to move from one framework to another, so they will need a sufficient incentive for this to be worthwhile. Both of these calculations are quantitative determinations - if a new framework only provides a small gain, that would likely be insufficient.
> > > >
> > > > **Our paper presents an evaluation framework at the beginning of a time when hardware and software specialization is growing in importance, and thus where comparative evaluations will become more important.** The economics of chip specialization have dramatically changed over the last decade or so (Thompson et al. 2021), leading Hennessy and Patterson to term this a “new golden age for computer architecture” in their Turing lecture (Hennessy and Patterson, 2019). But such specialization carries with it radical changes in performance. Comparing similar generations of GPUs and TPUs shows that TPUs get roughly a 6x performance increase in 16bit flop/s/1M transistors over GPUs, but that trying to do 32-bit operations can more than undo this benefit (Bjork et al. 2021). As specialization continues, disparities will only increase, as will the importance of co-designing implementations to those chips. Thus, we should expect that the type of quantitative portability analyses that we do in our paper will only become more important in the coming years.
> > > >
> > > > We hope our discussion and response to **fKz5** have adequately addressed the mentioned weaknesses perceived by **fKz5**. **If there are discussion points left that would prevent the reviewer from raising their score, please let us know, and we will further address them.**

---

> > > > > ### Comment · Reviewer_fKz5 · 2023-08-21
> > > > >
> > > > > Thanks for your response!
> > > > >
> > > > > There is no doubt that the authors put a lot of work into this project, and they should be appreciated for their efforts. However, the paper should be carefully revised to meet the standards of the conference. It is definitely important to study the solutions to the lack of portability, but the paper does not contribute much by its quantitive experiments without deeper analysis for application.
> > > > >
> > > > > The claim "because there has been a lack of quantitative evaluation, users of these frameworks needed to make the best implementation" cannot be supported by the quantitative results. In the manuscript, the author lists four of their contributions, but none of these address this challenge.
> > > > >
> > > > > I will raise my score to a borderline but my concerns maintain.

---

> > > > > > ### Author Response · Authors · 2023-08-21
> > > > > >
> > > > > > We thank **fKz5** for their valuable feedback and for updating their score to reflect the updates we made during the rebuttal period. We appreciate the fruitful discussion that has been had, and are very committed to updating the final manuscript to reflect the feedback and discussion. We are happy to engage in any further discussion to address any remaining concerns that might prevent the author further raising their score.

---

### Official Review · Reviewer_Twor · 2023-07-07

**Soundness:** 3 good
**Presentation:** 4 excellent
**Contribution:** 3 good
**Rating:** 6
**Confidence:** 4

**Summary:**

This paper studies the portability of three different ML frameworks (PyTorch, TensorFlow, and JAX) across different hardware types (GPUs and TPUs). The authors sample a variety of functions implemented in these frameworks and test to what extent these functions are interoperable across platforms, and also compare the function latency between the hardware types. The evaluation finds that many functions are either unimplemented, fail to execute, or execute far more slowly on one hardware type versus the other. For example, the authors report that JAX functions are almost always faster on TPUs as opposed to GPUs.

**Strengths:**

- In general, portability is a key practical concern and therefore the paper provides a valuable contribution for the ML community.
- Methodology for selecting functions for evaluation is comprehensive and the benchmarking is carefully executed.
- Thorough evaluation of GPU vs TPU execution reveals many interesting insights about function performance across hardware types.
- Paper includes granular analysis of failure cases.


**Weaknesses:**

- While portability is a very relevant and practical concern for ML practitioners, this paper only explores portability in the dimension of GPU vs TPU execution within the same framework. Another related dimension is how well functions (and more broadly models) map from one framework to another; for example, if I have a convolution kernel written in PyTorch executing on a GPU, how comparable is this kernel’s execution to a similar convolution operation in TensorFlow? While frameworks may share underlying cuDNN kernel implementations under the hood, it would be interesting to see to what extent this symmetry exists (it appears that some basic version of this analysis could already be performed using the data in Tables 4, 5, and 6). An extension of this dimension is how well frameworks operate with intermediate representation (IR) libraries like ONNX and OpenXLA. In particular, a helpful experiment might be taking a list of functions in PyTorch, exporting them to an IR and then exporting that IR to TensorFlow, and seeing what percentage of functions fail to execute, produce different outputs, and/or experience slowdowns.
- The paper only evaluates two specific hardware types: T4 GPUs and v3-8 TPUs. While the evaluation revealed significant disparities even between these platforms, the analysis would be even more compelling if expanded to include additional GPU / TPU types or new hardware types (e.g. AMD and/or Cerebras machines).


**Questions:**

- Why are certain functions so much slower on TPUs than GPUs and vice versa? This question may be difficult to answer given some kernel implementations are black box, but for kernels where source code is available it would be helpful to understand the performance behavior in more detail. For example, do certain kernels more effectively use hardware caches on GPUs vs TPUs?


**Limitations:**

The paper does not discuss limitations in much detail, but does discuss broader impacts. I do not foresee any negative societal impacts of this work.

---

> ### Author Rebuttal · Authors · 2023-08-10
>
> # Rebuttal (Twor)
>
> We thank reviewer Twor for their feedback and for noting the core strengths of our work, including "portability is a key practical concern and therefore the paper provides a valuable contribution for the ML community" and "evaluation of GPU vs TPU execution reveals many interesting insights about function performance across hardware types." We appreciate them mentioning the importance of portability. We believe portability is of extreme importance, and quantifying the lack of portability is a valuable contribution. We also appreciate review 2 highlighting the comprehensiveness of our methodology and benchmarking: "Methodology for selecting functions for evaluation is comprehensive and the benchmarking is carefully executed." Specifically, the reviewer mentioned our comparison of operations on GPU vs TPU and the granular categorization of failure modes.
>
> From reviewer Twor, we are addressing the following claims:
>
> 1. **"While portability is a very relevant and practical concern for ML practitioners, this paper only explores portability in the dimension of GPU vs TPU execution within the same framework. Another related dimension is how well functions (and more broadly models) map from one framework to another; for example, if I have a convolution kernel written in PyTorch executing on a GPU, how comparable is this kernel's execution to a similar convolution operation in TensorFlow?"**: We choose to focus our effort on hardware portability instead of framework portability because we believe it is the larger, more relevant problem. Software, and specifically frameworks, have no physical limitations on usage. While it may take the developer time to switch from framework to framework, this is a question of bits. Hardware is in the physical world. And as such, you are often limited by the devices you own or the amount of budget you have to access. While it would be fascinating to dive into how frameworks map to the underlying operations in CUDA and other compilation targets, we believe the first step was our process to find latencies and failure rates within a framework. Between the experimental cost of time and the compelling narrative these results gave, we believe this was a sufficient step to release this paper. We believe this is a promising direction for future research and will add it to our limitations and future work section in the manuscript.
> 2. **"An extension of this dimension is how well frameworks operate with intermediate representation (IR) libraries like ONNX and OpenXLA. In particular, a helpful experiment might be taking a list of functions in PyTorch, exporting them to an IR and then exporting that IR to TensorFlow, and seeing what percentage of functions fail to execute, produce different outputs, and/or experience slowdowns."**: While we agree exploring compilation targets and intermediate representations is a useful direction for this research, we believe it is a rich direction of future work. Here, our primary goal is to provide a primary rigorous contribution to evaluate the portability as a preliminary step to motivate such future work. Our experimental setup is already costly, requiring hand coding verifying, and running in a total of 1,146 experiments.
> 3. **"The paper only evaluates two specific hardware types: T4 GPUs and v3-8 TPUs. While the evaluation revealed significant disparities even between these platforms, the analysis would be even more compelling if expanded to include additional GPU / TPU types or new hardware types (e.g. AMD and/or Cerebras machines)."**: Agreed that more device types make for a stronger narrative when it comes to our results. We have taken the rebuttal period to incorporate the feedback of the reviewer and run additional experiments. In the attached pdf, we include a table showing failure rates for both the A100s and the T4s. Please see Table 1 in the attached file for the evaluation of A100s.
> 4. **"Why are certain functions so much slower on TPUs than GPUs and vice versa?"**: We thank the reviewer for the interesting question. To deeply understand why specific functions perform differently on TPUs and GPUs, we would need a granular profile of the operations down to the hardware level. However, in some cases, we do not have access due to the closed-source nature of CUDA and XLA code. Given that and the time cost of doing our current experiments, we believe that understanding why certain latency differences exist is beyond the scope of this paper. That said, even if the exact reasons for these lags are not fully investigated due to proprietary software barriers, recognizing their existence is crucial for the ML community.

---

> > ### Comment · Reviewer_Twor · 2023-08-13
> >
> > Thank you for the thoughtful response to the review and adding the additional A100 GPU results. I think the justification provided for the focus on hardware portability makes sense, though I would still be interested to see follow-up work in a subsequent paper on framework portability, as well. I do still think that some discussion about why certain functions are slower on each hardware type would strengthen the paper; even if there is no scope for running specific experiments for this paper, I would like to see some insight into A) possible explanations for this behavior and B) a high-level methodology for investigating the behavior in more detail (a single paragraph would suffice in my opinion). Overall, given the additional experiments added and the rebuttal text I have improved my score to a 6.

---

> > > ### Author Response · Authors · 2023-08-14
> > >
> > > We thank Twor for updating their score positively to reflect the additional experiments added and our clarifications in the rebuttal text. We welcome the feedback on providing additional observations to the reader in the discussion section as to why certain functions are slower on different hardware types. We will reply shortly before the end of this discussion phase with some additional context on A) and B) that we would be happy to include in the discussion and future works section of the final manuscript. We thank Twor again and are happy to engage further during the discussion period if any other questions come up.

---

> > > > ### Author Response · Authors · 2023-08-17
> > > >
> > > > We thank Twor again for their positive updates in this discussion stage. We take this opportunity to address one of their remaining requests to provide qualitative context for our empirical results that demonstrate certain functions show marked slowdowns when transferred from one hardware type to another.
> > > >
> > > > We include below some qualitative reasons we might see a performance difference between GPUs and TPUs and are happy to include this in the discussion section of the final manuscript. These are high-level and would need more exploration to say with certainty, but these hopefully provide some valuable context for readers for why the quantitative performance disparities we report. Broadly, we expect these performance differences to be attributed to one of the following categories:
> > > > 1. Misalignment between workload and implementation: frameworks and hardware may assume certain usage patterns (e.g., uniform input sizes) that are mismatched with actual workloads.
> > > > 2. Memory architectures: The substantially different memory architecture choices made by TPU and GPU architects advantage particular data structures and operations, making framework optimizations uneven in their effectiveness.
> > > > 3. Bottlenecks: Unimplemented features in some frameworks can lead to data transfer bottlenecks that lead to the full performance not being able to be taken advantage of.
> > > >
> > > > More specifically, we expect the following to be influencing our results:
> > > >
> > > > ## Pytorch
> > > > * Transfer of data from XLA to CPU or vice-versa sometimes takes a long time. This seems to be a problem that is much worse in PyTorch than TensorFlow due to a lack of an infeed processing implementation. TensorFlow specifically runs input processing on the CPU while TPU operations take place. As far as we can tell, PyTorch has chosen not to implement this, which makes TPUs slower than GPUs.
> > > > * Some operations may recompile their kernel when new shapes are seen when run upon TPUs. This can lead to slower results in tests when running on TPUs. Some of our tests may be using dynamic shapes, causing slowdowns.
> > > > ## Tensorflow
> > > > * Our testing shows a much larger difference between GPUs and TPUs for SVD operations than other functions. This is likely because the XLA program must be compiled for each input size for the function, requiring multiple compilations and thus, longer times on the TPU.
> > > > * While input processing is implemented in TensorFlow, data transfer remains a bottleneck. In some cases, this data preparation and transfer take longer than the XLA process itself.
> > > > The largest benefits of TPUs will be on operations involving matrix multiplication. For other operations, large speed-ups are not ensured.
> > > > TPUs generally work better with large batch sizes at multiples of 64. At other sizes, we may not see the full benefits of TPUs.
> > > >
> > > > ## Jax
> > > > - The performance difference in Jax between GPUs and TPUs is very small. It is very telling that a newer framework that was made with portability in mind works much more efficiently between devices.
> > > >
> > > > For future work to explore the quantitative impact of these differences in latency and uncover others, we would propose using more detailed profiling tools alongside our tests. This would allow us to observe specific memory usage, execution routing, etc.  Such an effort would be substantial since it would require both using multiple profiling tools (e.g., VTune, NSight, and Cloud TPU Tool) and establishing common baselines between them. Some granular insights may not be feasible due to the fact that both XLA and CUDA compilers are not open-source.
> > > >
> > > > We thank Twor again for the feedback which prompted these improvements to the discussion providing additional context to our quantitative results.

---

### Official Review · Reviewer_FGUA · 2023-07-07

**Soundness:** 4 excellent
**Presentation:** 3 good
**Contribution:** 3 good
**Rating:** 8
**Confidence:** 4

**Summary:**

The authors study the portability of a core set of PyTorch, TensorFlow, and JAX functions across hardware platforms, specifically GPUs and TPUs. They find significant fractions of functions fail to run on a given platform fully, partially, or within a tolerable latency. They provide the generated dataset.

**Strengths:**

- Timely analysis of portability of widely used ML software frameworks across different hardware devices
- Significant and surprising results that can go a long way toward shining a light on the issue and improving the current situation
- Well-written and clear exposition
- The public dataset release is highly beneficial for the community.

**Weaknesses:**

- For reproducibility, it would be useful to detail the exact software versions used for the benchmarking explicitly
- No CPU comparisons are made, which would be quite interesting to add given that many ML applications are run on CPUs (both x86 and ARM)
- In related work, there could be some discussion of the emerging ideas around FAIR for ML (https://www.rd-alliance.org/groups/fair-machine-learning-fair4ml-ig) and FAIR for research software (https://www.rd-alliance.org/groups/fair-research-software-fair4rs-wg)

**Questions:**

- What versions of the software were used?
- Is it possible to add comparisons to CPU (x86 or ARM)?

**Limitations:**

Yes, the authors have addressed limitations and broader societal impacts.

---

> ### Author Rebuttal · Authors · 2023-08-10
>
> # Rebuttal (FGUA)
>
> We thank the reviewer for their review and constructive feedback. We appreciate your recognition of the timely importance of our analysis of the portability of ML software frameworks across distinct hardware devices. It is great that you found our paper provides “Significant and surprising results that can go a long way toward shining a light on the issue and improving the current situation.” These results, as you pointed out, expose the existing challenges in the domain. Your acknowledgment of our “Well-written and clear exposition” and “The public dataset release is highly beneficial for the community” motivates us to continue this path.
>
> From reviewer FGUA, we are rebutting the following claims:
>
> 1. **"What versions of the software were used"**: We clarify that the Framework’s version: TensorFlow version 2.11.0, PyTorch version 1.12.0, and (3) JAX version 0.4.8. We will add these details to our paper, and thank the reviewer for pointing out this inadvertent omission.
> 2. **"Is it possible to add comparisons to CPU (x86 or ARM)?"**: Unfortunately, we were unable to complete these during the time allowed for the rebuttal – we prioritized the additional hardware variants on GPU since it takes time to run all functions and variants. However, we are happy to commit to adding these experiments to the final manuscript and agree that this is an important variant.
> 3. **"In related work, there could be some discussion of the emerging ideas around FAIR for ML (https://www.rd-alliance.org/groups/fair-machine-learning-fair4ml-ig) and FAIR for research software (https://www.rd-alliance.org/groups/fair-research-software-fair4rs-wg)"**: We thank the reviewer for the suggestions for a wider body of work that would be good to highlight. The two suggestions FAIR for ML (https://www.rd-alliance.org/groups/fair-machine-learning-fair4ml-ig) and FAIR for research software (https://www.rd-alliance.org/groups/fair-research-software-fair4rs-wg) relate to open science initiatives and multi-institutional collaborations to improve findability, accessibility, interoperability, and reuse of research software. These are important motivators of our current work. We will add these to the final manuscript as adjacent directions to understand the role of software in ensuring fairness.

---

> > ### Comment · Reviewer_FGUA · 2023-08-21
> >
> > Thank you for the response. While I feel a similar study for CPUs would be valuable, I acknowledge that given time and space constraints, that may be for a different paper. Overall, I maintain my original score.

---

> > > ### Author Response · Authors · 2023-08-21
> > >
> > > We thank **FGUA** for the time and effort put into your review. We appreciate the confidence in the merits of our work and thank you for the feedback during the rebuttal period.

---

### Author Rebuttal · Authors · 2023-08-10

# Global Response

We thank all reviewers for taking the time to evaluate our manuscript. We appreciate the recognition of the timeliness and relevance of our study (R fKz5 and FGUA), emphasizing the often overlooked issue of ML software framework portability across different hardware types (R fKz5). It is encouraging to see the consensus on the value of our large-scale experimental results, which can serve as a reference for both academia and industry (R fKz5, Twor, FGUA). The thorough and careful evaluation methodologies employed, particularly our comprehensive approach to function selection and benchmarking (R Twor), as well as our granular analysis of failure cases (R Twor), are recognized positively. Furthermore, we are grateful for acknowledging our contribution in revealing significant insights about function performance across GPUs and TPUs (R FGUA, Twor).

As reviewer (Twor) has suggested we have included a pdf with results for another type of GPUs. We now include both T4 and A100 GPU results. We believe this makes our research even stronger and we hope to extend to more types of devices in the future.

---

### Decision · Program_Chairs · 2023-09-21

**Decision:**

Accept (poster)

**Comment:**

I have some reservations to accept this paper but given the overall support from the reviewers, I recommend acceptance for the importance of the topic which is not discussed in ML conferences enough.

My main reservation is that the results are not very surprising. Jax was built on a solid foundation of XLA. Thus it is more portable than other frameworks by design. 99 % of the PyTorch users do not have access to TPUs. Thus it is not surprising that the operations are not supported on TPUs. TensorFlow's lack of portability is a bit disappointing. One can only speculate that this must be a consequence of Google's internal resourcing decisions.

On the methodology, I have the following questions

1. Does it make sense that the top 20 functions are chosen by the frequency of usage rather than their impact to some representative model's runtime? For example, certainly `torch.cuda` is used a lot but I don't expect this function to be a bottleneck.
2. Does it make sense that the functions are sampled independently for each framework? For example, `svd` is included in tensorflow and jax but `torch.svd` is not. I think the authors should benchmark important functions for all frameworks.
3. Does it make sense that the authors benchmark the unit tests? Inputs to unit tests are usually small and the performance on them can be very different from the performance on typical inputs to these functions in a real model.